# Dual lineage origins contribute to neocortical astrocyte diversity

Jiafeng Zhou ®[1], Ilaria Vitali ®[2], Sergi Roig-Puiggros ®[1], Awais Javed[1], Iva Cantando[3], Matteo Puglisi ®[4,5], Paola Bezzi ®[3,6], Denis Jabaudon ®[1], Christian Mayer ®[2] & Riccardo Bocchi ®[1] ✉

Astrocytes are not a uniform population but exhibit diverse morphological, molecular, and functional characteristics. However, how this diversity originates and becomes establishes during development, remains largely unknown. Here, using single-cell RNA sequencing and spatial transcriptomics, we identify five astrocyte subtypes with unique molecular features, spatial distributions and functions in the mouse neocortex and characterize essential regulators for their formation. Using TrackerSeq to trace clonally related astrocytes, we identify two distinct lineages that give rise to these five subtypes. One lineage derives from *Emx1*+ radial glial cells that initially generate neurons and later switch to astrocyte production. The other, with minimal neuronal output, predominantly produces a distinct subset of astrocytes marked by Olig2. *Olig2* knockout disrupts lineage specification, leading to changes at molecular, morphological and functional levels. These findings shed light on the cellular mechanisms underlying astrocyte diversity, highlighting the presence of multiple radial glial cell subtypes responsible for generating cortical astrocyte subtypes.

Astrocytes are a crucial cell type in the mammalian brain, serving a wide range of physiological functions, including maintenance of neuron viability, formation of the blood brain barrier, and regulation of synapse functions[1]. Although traditionally viewed as a relatively homogeneous population, recent studies have revealed that astrocyte subtypes displaying distinct molecular features are present throughout the central nervous system[2–6]. Additionally, some intra-regional molecular differences have been detected within the neocortex[7,8]. Despite these advances, there is still no consensus on the extent of heterogeneity among cortical astrocytes. More critically, the developmental mechanisms underlying their diversity remain largely unknown. Furthermore, the link between the molecular subtypes of astrocytes and their specific functions has yet to be fully elucidated.

To address these questions, we employed a cross-modal approach combining high-throughput single-cell RNA sequencing (scRNA-seq) and Multiplexed Error-Robust Fluorescence in situ Hybridization (MERFISH). These methods allowed us to identify five molecularly distinct astrocyte subtypes in the mouse neocortex, each with unique localizations along the cortical column and distinct predicted functions. Taking advantage of a diffusion-based computational approach (URD)[9], a method to simulate astrocyte development, we identified key regulators underlying the formation of each subtype. By combining scRNA-seq and massively parallel clonal-tagging (TrackerSeq)[10] of radial glial cells (RGCs), we traced clonally related cortical astrocytes and showed that astrocyte subtypes arise from two separate lineages, i.e., distinct RGCs. In addition to the previously described *Emx1*+ RGC capable of switching their potency to produce neurons first and then

[1]Department of Basic Neurosciences, University of Geneva, Geneva, Switzerland. [2]Max Planck Institute for Biological Intelligence, Martinsried, Germany. [3]Department of Fundamental Neurosciences, University of Lausanne, Lausanne, Switzerland. [4]Division of Physiological Genomics, Biomedical Center, Ludwig-Maximilians-Universität München, Planegg-Martinsried, Germany. [5]Institute for Stem Cell Research, Helmholtz Zentrum München Deutsches Forschungszentrum für Gesundheit und Umwelt (GmbH), Nuremberg, Germany. [6]Department of Physiology and Pharmacology, Sapienza University of Rome, Rome, Italy. ✉e-mail: riccardo.bocchi@unige.ch

astrocytes[11], we identified a RGC population already present at early stages of the corticogenesis. This population, displays limited neuronal output and gives rise predominantly to a distinct subset of astrocytes expressing the transcription factor Olig2. These findings provide an alternative view to the current model, wherein a single population of multipotent RGCs generate virtually all cortical astrocytes[11], transitioning from neurogenesis to gliogenesis[12]. Our results demonstrated the existence of two distinct RGC subtypes within the ventricular zone, coexisting and jointly contributing to the generation of cortical astrocytes diversity.

## Results

### Molecular identity and spatial distribution reveal five cortical astrocyte subtypes

To investigate the molecular heterogeneity of cortical astrocytes, we generated scRNA-seq datasets from wild-type mouse cortices at postnatal day (P)7, a stage when astrocytes delamination and expansion are complete[13]. Electroporation of episomal plasmids does not allow permanent labeling of astrocytes, due to their high division rate and consequent dilution of the plasmid[14]. Therefore, we employed a piggyBac transposase-based system[15] that integrate the sequence of the green fluorescent protein (GFP) from a donor plasmid into the genome of *in utero* electroporated dorsal RGCs lining the lateral ventricle (Fig. 1a, Supplementary Fig. 1a). To ensure the inclusion of all astrocytes subpopulations, we targeted RGCs at embryonic day (E)12.5 or E16.5, and collected GFP+ cells at P7 using fluorescence-activated cell sorting (FACS) for scRNA-seq (Fig. 1a, Supplementary Fig. 2). These datasets were integrated with a comprehensive transcriptomic atlas of the developing neocortex spanning from E10.5 to P4[16], to facilitate cell type annotations and the identification of the maturation stages (Supplementary Fig. 1b). By employing established gene markers for well-characterized cell populations (Supplementary Data 1), we calculated cell type scores and annotated all known cell types in the neocortex, including astrocytes (Supplementary Fig. 1c–e). Astrocyte and oligodendrocytes/oligodendrocyte precursor cells (OPCs) predominated in the P7 dataset, while excitatory neurons and RGCs were primarily detected at earlier stages (Supplementary Fig. 1f). We then subset astrocytes and re-clustered them for subtype analysis (Fig. 1b). We identify five distinct astrocyte subtypes, characterized by unique molecular markers (Fig. 1c, d and Supplementary Data 2). To predict their cortical locations, we generated two MERFISH spatial transcriptomics datasets from coronal sections of P7 and adult mouse brains (Supplementary Fig. 3a). These datasets include 457 representative genes (Supplementary Data 3), enabling the identification of all known cell types (Supplementary Fig. 3b). By isolating the astrocyte cluster (Supplementary Fig. 3b, c), we mapped scRNA-seq-defined subtypes onto MERFISH brain sections. At both P7 and adulthood, astrocyte subtypes exhibited similar spatial distributions, occupying distinct positions along the cortical column (Fig. 1e, f). Although slight differences were observed between the two stages, likely reflecting developmental processes still underway at P7, their overall organization was already fairly well established by this stage. Specifically, Ast_Slc7a10 was enriched in the upper cortical gray matter (GM), Ast_S100a4 was localized to the white matter (WM), and Ast_Serpinf1 was concentrated in layer 1 and the lower GM, while Ast_H2az1 and Ast_Sfrp1 were more broadly distributed throughout the GM. Similar spatial distributions were observed when mapping astrocyte subtypes onto our previously published adult mouse Visium dataset (Supplementary Fig. 3d)[17,18]. To confirm their persistence, we integrated two adult datasets[19,20] and transferred P7 subtype labels, identifying all five subtypes in the adult brain (Supplementary Fig. 3e). Notably, while these subtypes remained consistent across stages, certain marker expressions varied between P7 and adult MERFISH datasets, likely reflecting maturation-related molecular changes (Supplementary Fig. 3f)[19,20]. These findings indicate that cortical astrocytes form a heterogeneous population of five spatially organized subtypes.

### Two distinct lineages underlie cortical astrocyte diversity

To investigate the emergence of cortical astrocyte heterogeneity, we subset RGC and astrocyte populations from E10.5 to P7 from the integrated scRNA-seq datasets (Supplementary Fig. 1c) and re-clustered them (Fig. 2a). Cell type scores and developmental stages (Supplementary Fig. 4a, b) revealed a continuum from early RGCs to immature astrocytes and ultimately to distinct subtypes (Fig. 2a). Pseudotime analysis revealed a temporal gradient from early to late stages, indicating a progressive differentiation from RGCs to astrocytes (Fig. 2b). Using a diffusion-based computational approach (URD)[9], which reconstructed a branched tree structure based on the transcriptional similarity and pseudotime, we identified a key branching point around E16.5, where two major trajectories, named S100a11 and Olig2, gave rise to the five astrocyte subtypes (Fig. 2c). These trajectories showed distinct spatial distributions across P7 MERFISH, adult MERFISH, and adult Visium datasets: Olig2-derived astrocytes enriched in cortical GM and layer 1; S100a11-derived astrocytes predominantly in WM with scattered GM presence (Supplementary Fig. 4c, d). Mapping transcriptional dynamics along the differentiation trajectories, we observed a sequential downregulation of RGC genes (*e.g., Pax6, Sox2, Hes1*), followed by the upregulation of pan-astrocytic genes (*e.g., Slc1a3, Sox9, Clu*) in both trajectories, albeit with differential temporal patterns (Fig. 2d). The Olig2 trajectory exhibited an earlier downregulation of RGC genes, along with an earlier expression of pan-astrocytic genes, compared to the S100a11 trajectory. Subtype composition across development stages confirmed that Olig2-derived astrocytes emerged before S100a11-derived ones (Supplementary Fig. 4e), suggesting that astrogenesis begins earlier in the Olig2 trajectory than in the S100a11 trajectory. At later stages, we identified two distinct sets of genes that were selectively enriched in either the S100a11 trajectory (*e.g., S100a11, Riiad1*, and *Phlda1*), or the Olig2 trajectory (*e.g., Olig2, Chrdl1*, and *Egfr*; Fig. 2d, e, Supplementary Fig. 4f). To validate the trajectory analysis, we calculated trajectory scores based on these two gene sets (Supplementary Data 4). The Olig2 trajectory score was higher in Ast_Slc7a10, Ast_Serpinf1 and Ast_H2az1, while the S100a11 trajectory score demarcated the Ast_Sfrp1 and Ast_S100a4 subtypes (Fig. 2f). Collectively, our molecular analysis identifies two principal gene cascades underlying distinct astrocyte trajectories, underscoring their crucial role in shaping the molecular heterogeneity of cortical astrocytes.

To further explore the evolutionary significance of these two distinct astrocyte trajectories, we analyzed a publicly available scRNA-seq dataset from the developing human neocortex[21]. Our findings confirmed that both molecular astrocyte trajectories are present in the human neocortex (Supplementary Fig. 5a–d), supporting their conservation across mammalian species. In contrast, non-mammalian species, such as reptiles[22] and birds[23], revealed that their astrocytes predominately belong to the Olig2 trajectory (Supplementary Fig. 5e–g). This suggests that the diversification of astrocyte trajectories represents an evolutionary advancement specific to the mammalian neocortex, potentially contributing to its enhanced functional complexity.

To access the contribution of clonality to astrocytes trajectories formation, we used TrackerSeq[10], a high throughput tagging system based on the piggyBac transposase, which permanently integrates a GFP reporter gene along with a partially random synthetic oligonucleotide sequence (*i.e.*, barcode), detectable through scRNA-seq (Fig. 2g, top). We co-electroporated the helper and donor plasmids at E12.5, and FAC-Sorted GFP+ cells at P7 for scRNA-seq analysis (Fig. 2g, bottom). The dataset captured all major neocortical cell types, with

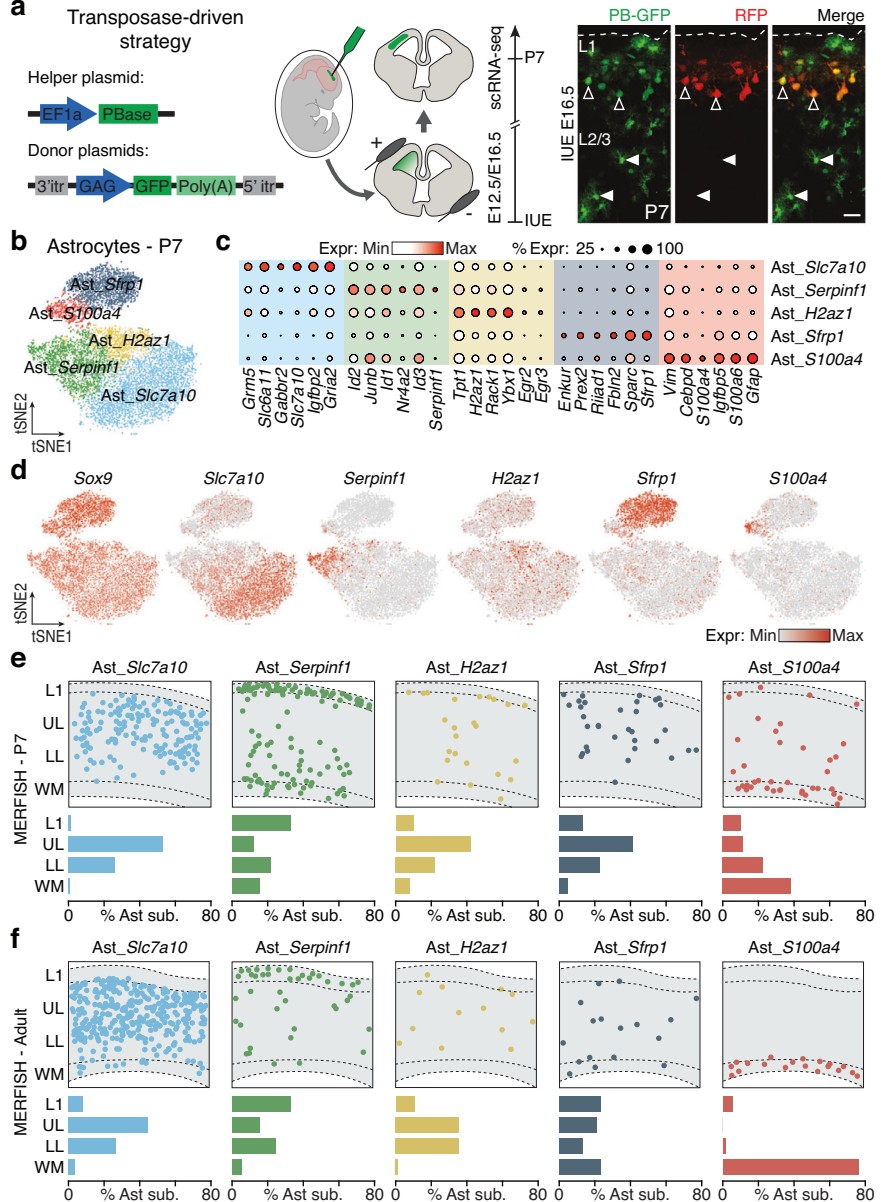

**Fig. 1 | Identification and spatial distribution of astrocyte subtypes in the mouse neocortex. a** Left: schematic of the plasmids and of the *in utero* electroporation (IUE) technique employed to label cortical astrocytes. Right: upper part of a P7 cortex electroporated at E16.5 with integrative (PB-GFP) and episomal (RFP) plasmids. Empty arrowheads indicate GFP⁺/RFP⁺ neurons while full arrowheads indicate GFP⁺/RFP⁻ astrocytes. Scale bar: 50 µm. **b** tSNE representation of P7 astrocyte subtypes. **c** Expression levels of representative genes for each identified astrocyte subtypes (full gene list in Supplementary Data 2). **d** tSNE plots illustrating expression levels of selected genes across astrocyte subtypes with *Sox9* as pan-astrocytic marker. **e, f** Spatial positioning of astrocyte subtypes (top) and their distribution along the cortical column (bottom) in P7 ((**e**) cropped from Supplementary Fig. 3a top) and adult ((**f**) cropped from Supplementary Fig. 3a bottom) MERFISH coronal sections. IUE: *in utero* electroporation; L1: layer 1; UL: upper layer; LL: lower layer; WM: white matter.

75% comprising excitatory neurons, astrocytes, and oligodendrocytes/OPCs (Supplementary Fig. 6a). Barcodes were recovered in approximately 75% of cells (Supplementary Fig. 6b), yielding 958 clones with an average size of 4.5 cells (Supplementary Fig. 6c). Some clones were shared across excitatory neurons, astrocytes, and oligodendrocytes/OPCs, while others were restricted to excitatory neurons, consistent with previous findings (Supplementary Fig. 6d, e)[11]. In contrast, a subset of clones was exclusively composed of glial cells (astrocyte and/or oligodendrocytes/OPCs), suggesting the existence of glia-enriched lineages. To probe clonal relationships, we performed hierarchical clustering of lineage coupling scores (measuring barcode sharing between cell types) and found a lineage cluster composed of RGCs, neuroblasts, and neurons (Supplementary Fig. 6f). In contrast,

astrocytes and oligodendrocytes/OPCs exhibited lower lineage coupling correlations with other cell types, reinforcing the concept of glia-enriched lineages (Supplementary Fig. 6f). Similar patterns emerged when analyzing clonal relationships among astrocyte, oligodendrocytes/OPCs, and excitatory neuron subtypes (Supplementary Fig. 6g). The lineage coupling analysis identified three main lineages: one corresponding to oligodendrocytes/OPCs subtypes, another comprising astrocyte subtypes, and a third mixed lineage containing of astrocytes, oligodendrocytes/OPCs, and excitatory neurons (Supplementary Fig. 6h).

To further explore clonal relationship between astrocyte subtypes and neurons, we subset and re-clustered astrocyte and neuron from the E12.5-P7 dataset (Fig. 2h). Clonal analysis revealed that S100a11

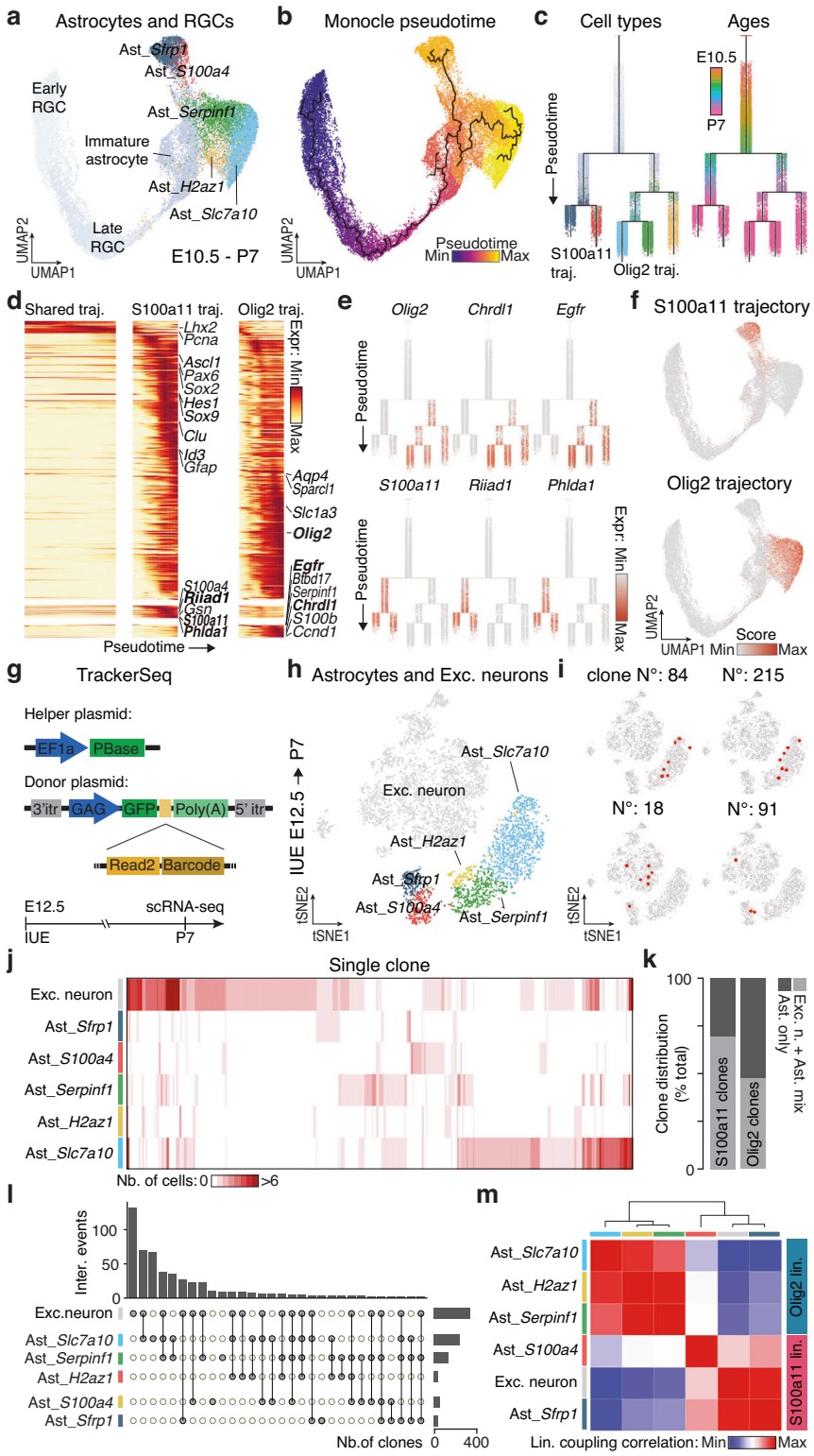

trajectory astrocytes (Ast_Sfrp1 and Ast_S100a4) frequently shared clones with excitatory neurons whereas Olig2 trajectory astrocytes (Ast_Slc7a10, Ast_Serpinf1, and Ast_H2az1; Fig. 2i, j) were more often restricted within the astrocyte lineage. Quantifications showed that ~75% of S100a11 trajectory clones were shared with excitatory neurons, compared to lower proportion in the Olig2 trajectory (Fig. 2k, l), suggesting a closer developmental relationship between S100a11

astrocytes and excitatory neurons. This is further supported by lineage coupling analysis which showed stronger correlations between S100a11 astrocytes and excitatory neurons (Fig. 2m), and weaker coupling for Olig2 astrocytes (Fig. 2m). Overall, these findings indicate that the five astrocyte subtypes arise from two distinct lineages: the S100a11 lineage, which generates a subset of astrocytes sharing a common origin with excitatory neurons, and the Olig2 lineage, which

**Fig. 2 | Cortical astrocyte subtypes originate from two distinct lineages.**
**a** UMAP representation of RGCs and astrocytes from E10.5 to P7. **b** UMAP representation of Monocle pseudotime underlying the developmental trajectory from RGCs to astrocyte subtypes. **c** URD trees simulating the development from RGCs to astrocyte subtypes, color coded by identities (left) or ages (right). **d** Gene expression cascades of differentially expressed genes between S100a11 and Olig2 trajectories, ordered by temporal dynamics and divided into three segments: shared trajectory, S100a11 trajectory (left clade) and Olig2 trajectory (right clade). Selected genes are labeled (full gene list in Supplementary Data 4). **e** Expression of trajectory-specific genes along URD tree. **f** UMAP plots of S100a11 and Olig2 trajectory scores, based on trajectory-specific genes. **g** Schematic of the TrackerSeq plasmids and workflow. **h** tSNE representation of excitatory neurons and astrocyte subtypes traced from E12.5 and collected at P7. **i** Examples of clones that are either restricted within Olig2 trajectory astrocytes (top) or shared between excitatory neurons and S100a11 trajectory astrocytes (bottom). **j** Clonal distributions across excitatory neurons and astrocyte subtypes, with each column representing a clone and cell numbers indicated by the color scale. **k** Proportion of clones with only astrocytes (Ast. only) or both astrocytes and excitatory neurons (Exc. n. + Ast. mix). **l** UpSet plot showing the number of clones shared or unique between excitatory neurons and astrocyte subtypes. Top bar graphs indicate observed intersections; right bar graphs show clone counts per cell subtype. **m** Heatmap of lineage coupling scores between excitatory neurons and astrocyte subtypes, from positive (red, coupled) to negative (blue, anti-coupled). IUE: *in utero* electroporation.

predominantly generates the remaining astrocyte subtypes with minimal neuronal contribution.

## Two RGC subtypes account for cortical astrocyte diversity

To identify RGC subtypes giving rise the two astrocyte lineages, we traced E12.5 RGCs with TrackerSeq[10] and collected their progenies for scRNA-seq at E18.5 to capture the molecular profiles of various cellular states of RGCs and astrocytes, within a single comprehensive dataset (Fig. 3a, top). The dataset included RGCs, OPCs, excitatory neurons, and both astrocyte lineages (Fig. 3b, Supplementary Fig. 7a). Using Di Bella et al. dataset[16] as a reference (Fig. 3a, bottom), we identified two RGC subtypes: RGC_1 and RGC_2 (Fig. 3b). Clonal analysis revealed that RGC_1 shared clones with S100a11 astrocytes and excitatory neurons, whereas RGC_2 clones were largely restricted to Olig2 lineage astrocytes (Supplementary Fig. 7b). Specifically, 50% of RGC_2 clones were within the Olig2 lineage, 34% self-renewing, and 16% linked to neurons, but none to S100a11 astrocytes (Fig. 3c). In contrast, 90% of RGC_1 clones were linked to neurons and 5% each to Olig2 and S100a11 astrocytes (Fig. 3c). Lineage coupling analysis confirmed RGC_1's relationship with neurons and S100a11 astrocytes, while RGC_2 was more closely linked to Olig2 astrocytes (Supplementary Fig. 7c). Restricting the analysis to RGC_1, RGC_2, neurons, and astrocytes further confirmed these distinct lineage relationships (Fig. 3d).

Differential gene expression analysis at E18.5 revealed distinct molecular signatures for the two RGC subtypes: RGC_1 expressed *Ascl1* and *Hes6*, while RGC_2 was marked by *Id3* and *Serpine2* (Fig. 3e, f, Supplementary Data 5). The proportion of RGC_1 decreased from 60% at E17 to 44% at P2, while RGC_2 increased from 36% to 52% (Fig. 3g, h), suggesting greater proliferative activity in RGC_2. This was confirmed by Ki67 staining at P0, with over 50% of RGC_2 proliferating versus ~30% of RGC_1 (Supplementary Fig. 7d, e). Cell cycle analysis showed >70% of RGC_2 in G2/M, compared <50% of RGC_1 (Supplementary Fig. 7f), and that Olig2 lineage clones contained more astrocytes per clone than those from the S100a11 lineage at P7 (Supplementary Fig. 7g). These findings highlight two RGC subtypes with distinct molecular profiles and proliferative dynamics contributing to gliogenesis.

To access RGC heterogeneity at earlier stages, we transferred the cell subtype labels from our E12.5-E18.5 dataset to a RGC dataset spanning E10.5 till E16.5, obtained from the Di Bella et al. (Supplementary Fig. 7h, left)[16]. While most cells were assigned as RGC_1, we identified a subset of cells labeled as RGC_2 as early as E10.5, indicating the existence of RGC_2 at the onset of corticogenesis (Supplementary Fig. 7h, right). To investigate the molecular changes of these two RGC subtypes during corticogenesis, we calculated the average PCA distance within each subtype at different ages to assess their transcriptional stability over time (Supplementary Fig. 7i). RGC_1 showed increasing transcriptional divergence[24], consistent with progressive cell state transitions during neurogenesis[12], whereas RGC_2 remained transcriptionally more stable. To further explore molecular divergence of RGC_1 and RGC_2 during corticogenesis, we applied the URD algorithm to the RGC dataset. The resulting branched structure revealed that divergence between RGC_1 and RGC_2 starts around E12.5 (Fig. 3i), marked by two distinct gene expression programs specific to each subtype (Fig. 3j, Supplementary Fig. 8a and Supplementary Data 5). Consistent with previous studies[25,26], we observed that RGC_1 highly expressed *Emx1* (Supplementary Fig. 8a), a well-known marker for excitatory neuronal lineage. To validate the existence and spatial distribution of these subtypes, we generated a MERFISH spatial transcriptomics dataset from an E14.5 mouse brain coronal section, profiling 457 genes (Supplementary Data 3), including those differentially expressed between RGC_1 and RGC_2 (Fig. 3j). After annotating this MERFISH dataset, we subset the RGC cluster (Supplementary Fig. 8b, c) and transferred RGC_1 and RGC_2 identities from the RGC dataset. This revealed two spatially distinct RGC populations (Fig. 3k), with *Emx1* and *Lhx2* enriched in RGC_1, and *Zbtb20* and *Ptprz1* in RGC_2 (Fig. 3l, Supplementary Fig. 8d).

Given the strong *Emx1* expression in RGC_1 and its low or absent expression in RGC_2 (Supplementary Fig. 8a), we crossed the *Emx1::cre* mouse line with the *ROSA[nT-nG]* nuclear reporter line[25,27], which labels Cre⁺ cells with GFP and Cre⁻ cells with RFP, to trace the progeny of *Emx1*⁺ RGCs. To validate its specificity, we assessed the proportions of excitatory and inhibitory neurons traced until P7. Nearly all Neurod2⁺ excitatory neurons were GFP⁺, and almost all GABA⁺ inhibitory neurons were RFP⁺ (Supplementary Fig. 9a, b), consistent with previous findings[25,26]. At E12.5, ~80% of cortical RGCs were GFP⁺ (Supplementary Fig. 9c, d), indicating that the *Emx1*⁺ RGC is not the sole RGC subtype present in the ventricular and subventricular zones. Although this proportion increase slightly over time, still ~10% of RGCs were GFP⁻/RFP⁺ at E16.5 (Supplementary Fig. 9e, f). These proportions align with those observed in the E14.5 MERFISH dataset, where most RGCs are labeled as RGC_1 and only a minority as RGC_2 (Fig. 3k). At P7, co-staining for Sox9 and Olig2 (identifying astrocytes from the Olig2 lineage, Fig. 3m) showed that ~30% of cortical astrocytes were GFP⁻/RFP⁺, with 80% expressing Olig2 (Fig. 3n). Similarly, S100b⁺ astrocyte quantification revealed comparable proportions of non-*Emx1*-derived astrocytes, with most also expressing Olig2 (Supplementary Fig. 9g). To exclude oligodendrocytes/OPCs contamination, we quantified Sox9⁺/Sox10⁻ astrocytes, revealing ~25% were GFP⁻/RFP⁺ and 75% GFP⁺/RFP⁻ (Supplementary Fig. 9h). Similar analysis, using the *Ai14* reporter mouse line, also revealed the presence of astrocytes not derived from *Emx1*⁺ RGCs. (Supplementary Fig. 9i)[28]. The observed proportions of GFP⁺ and RFP⁺ RGCs and astrocytes support the existence of two RGC subtypes, RGC_1 and RGC_2, giving rise to distinct astrocyte lineages, S100a11 and Olig2 lineages.

## Olig2 is the key regulator for fate specification of Olig2 lineage

Among the genes differentially expressed between the two astrocyte lineages, we identified Olig2 (Fig. 2d, e), a key transcription factor in astrocyte development[29–31]. Analysis of the E18.5 scRNA-seq dataset revealed that *Olig2* is initially expressed in both RGC subtypes but is rapidly downregulated as RGC_1 differentiates into S100a11 lineage

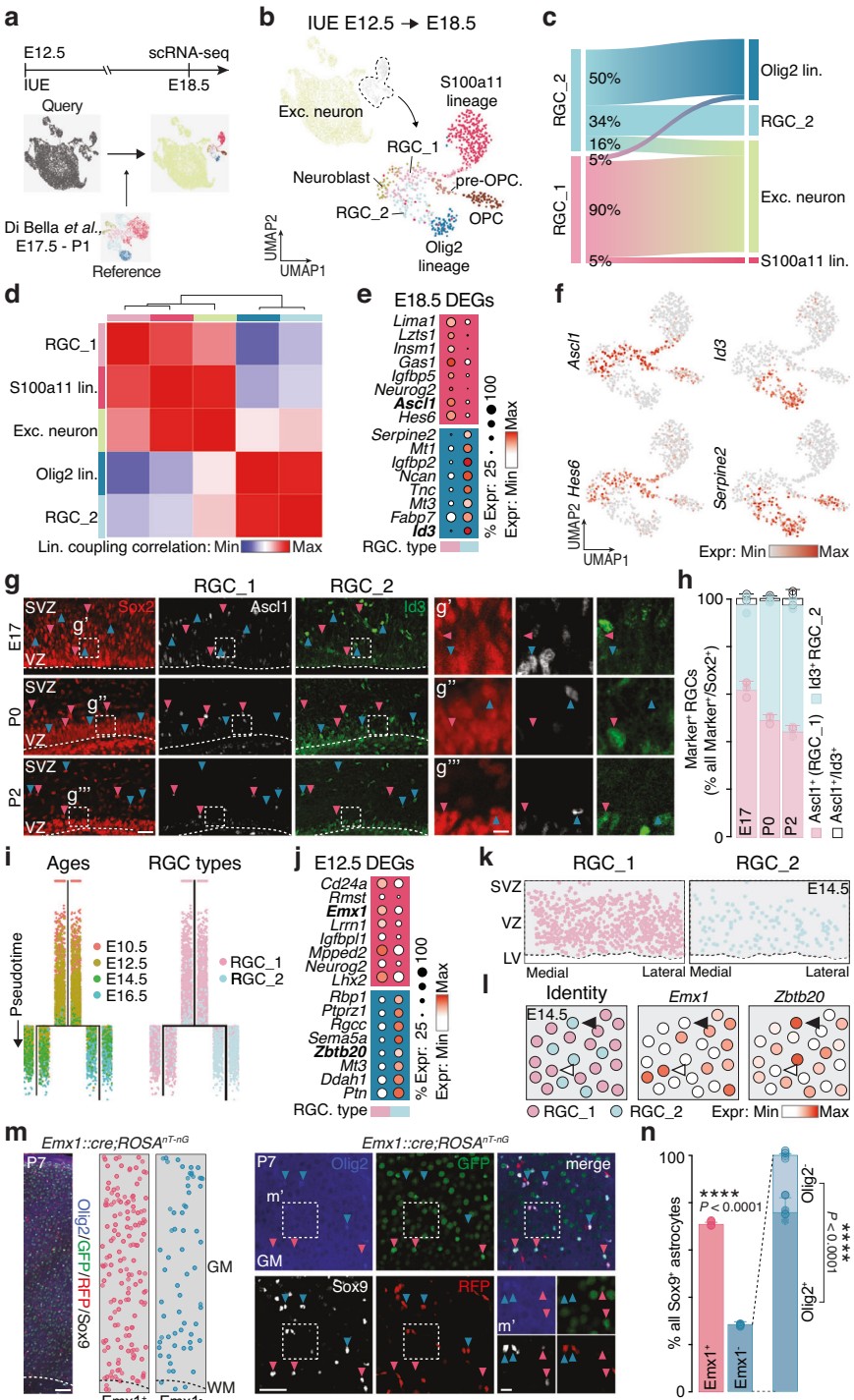

**Fig. 3 | Cortical astrocytes originate from two distinct RGC subtypes.**
**a** Experimental strategy for collecting and annotating the E18.5 scRNA-seq dataset.
**b** UMAP representation of E18.5 scRNA-seq dataset, colored by cell type.
**c** Proportion of RGC_1 and RGC_2 clones shared with RGCs, neuron, and astrocyte linages, indicating lineage relationships. **d** Heatmap of lineage coupling scores between E18.5 excitatory neurons, astrocyte lineages and RGC subtypes, from positive (red, coupled) to negative (blue, anti-coupled). **e** Expression of the top eight genes per RGC subtype at E18.5 (full gene list in Supplementary Data 5).
**f** UMAP plots illustrating expression of selected genes. **g** Ascl1 and Id3 staining in Sox2⁺ RGCs in the VZ/SVZ across developmental stages. Scale bars: 50 μm (left), 10 μm (right). **h** Fraction of Ascl1⁺ and Id3⁺ RGCs among Marker⁺/Sox2⁺ RGCs. $n = 3$ animals. **i** URD branching trees tracing RGC trajectories from E10.5 to E16.5, colored by developmental stage (left) or predicted identity (right). **j** Expression of the top eight genes for each RGC subtype at E12.5 (see Supplementary Data 5 for

the full gene list). **k** Spatial mapping of RGC subtypes onto an E14.5 MERFISH coronal section. **l** *Emx1* and *Zbtb20* expression in a cropped region of the E14.5 MERFISH dataset. Empty arrowheads: RGC_1 *Emx1⁺*/ *Zbtb20⁻*; full arrowheads: RGC_2 *Emx1⁻*/*Zbtb20⁺*. **m** Left: P7 cortex from *Emx1::cre;Rosa^{nT-nG}* transgenic mouse immunostained for Sox9 and Olig2. Each dot represents a Sox9⁺ astrocyte, color-coded by Emx1 expression. Right: examples of Sox9⁺ astrocytes: Sox9⁺/*Emx1⁺*/Olig2⁻ astrocytes are indicated by pink arrowheads while Sox9⁺/*Emx1⁻*/Olig2⁺ astrocytes by blue arrowheads. Scale bars: 100 μm (left), 50 μm (middle) and 10 μm (right). **n** Quantifications of Sox9⁺/*Emx1⁺* and Sox9⁺/*Emx1⁻* astrocytes in the entire cortical column. Olig2 expression is further quantified in all Sox9⁺/*Emx1⁻* astrocytes. $n = 5$ animals; two-tailed *t*-test. ****$p < 0.0001$. Values are shown as mean ± s.d.. IUE: *in utero* electroporation; WM: white matter; GM: gray matter; VZ: ventricular zone; SVZ: subventricular zone; LV: lateral ventricle. Source data are provided as a Source Data file.

astrocytes (Supplementary Fig. 10a, b, left). In contrast, its expression persists in Olig2 lineage astrocytes through the second postnatal week (Supplementary Fig. 10a, b, right)[29]. To validate this, we quantified Olig2[+] cells across RGCs and astrocyte lineages using Pax6, Ascl1, and Id3 as markers. Since Pax6 labels both RGCs and astrocytes[32], we classified Pax6[+] cells in the cortical plate (CP) as astrocytes and those below as RGCs. Within RGCs, Ascl1 marked RGC_1, and Id3 identified RGC_2. In the CP, Id3[+]/Pax6[+] cells represent Olig2 lineage astrocytes, and Id3[-]/Pax6[+] cells the S100a11 lineage astrocytes (Supplementary Fig. 10c, d). Quantification revealed that ~60% of RGC_1 initially expressed Olig2, dropping to <20% in S100a11 lineage astrocytes. In contrast, >50% of RGC_2 were Olig2[+], increasing to 80% in Olig2 lineage astrocytes (Supplementary Fig. 10e), supporting the expression pattern observed in the E18.5 scRNA-seq dataset.

To investigate the regulation of Olig2, we performed gene regulatory network (GRN) analysis at E18.5 using CellOracle (Supplementary Fig. 10f, left and Supplementary Data 6)[33]. This revealed both negative and positive regulators of Olig2 in RGCs (Supplementary Fig. 10f, center). Negative regulators (e.g., Fezf2, Plagl1) were enriched in S100a11 lineage astrocytes, possibly driving Olig2 downregulation, while positive regulators (e.g., Egr1, Tfdp1) were enriched in Olig2 lineage astrocytes, supporting its sustained expression (Supplementary Fig. 10g). GRN analysis also identified Olig2 target genes: negatively regulated genes (e.g., Thrsp and Trim2) were enriched in S100a11 lineage astrocytes, while positively regulated genes, (e.g., Olig1 and Serpine2) were prominent in Olig2 lineage astrocytes (Supplementary Fig. 10f, g). These findings suggest that Olig2 plays a crucial role in defining astrocyte lineage identity by integrating regulatory inputs from both upstream transcription factors and its downstream gene network.

To test functional role of Olig2 in astrocyte lineage specification, we performed a CRISPR-mediated knockout. We electroporated TrackerSeq at E16.5, along with a Cas9 plasmid and two Olig2-targeting gRNAs (Fig. 4a)[34]. At P7, Olig2 immunostaining showed a significant decrease in Olig2[+]/Sox9[+] (Fig. 4b, c, left) and Olig2[+]/S100b[+] astrocytes (Supplementary Fig. 11a, left), indicating disrupted generation of Olig2 lineage astrocytes. This was further supported by a decrease in Egfr[+] astrocytes, a marker of the Olig2 lineage (Supplementary Fig. 11b). Despite this reduction, we observed a slight increase in total Sox9[+] (Fig. 4c, right) and S100b[+] astrocytes (Supplementary Fig. 11a, right), in line with previous studies showing that Olig2 represses astrogenesis[35,36]. In addition, there was an increase in GFAP[+] astrocytes among electroporated cells, suggesting a shift toward the S100a11 lineage (Supplementary Fig. 11c). Spatial analysis showed that, under control conditions, Olig2[+] astrocytes were enriched in the GM, while Olig2[-] (S100a11) astrocytes localized to mainly the WM (Fig. 4d, left), consistent with spatial transcriptomic data (Fig. 1e, f; Supplementary Fig. 5c, d). Upon Olig2 knockout condition, Sox9[+] astrocytes exhibited an enrichment in the WM, mirroring the distribution of control Olig2[-] astrocytes (Fig. 4d). Morphologically, P7 control Olig2[+] astrocytes had a larger diameter and more complex branching than Olig2[-] astrocytes (Fig. 4e–g, Supplementary Fig. 11d). PCA of morphological features revealed that astrocytes from the control Olig2[+] group clustered separately from both control Olig2[-] and Olig2 knockout astrocytes, while the latter two groups were closer in PCA space (Fig. 4h). Similar trends were observed at P14, indicating persistent changes across conditions (Supplementary Fig. 11e–g). Together, these results show that Olig2 knockout astrocytes acquire the spatial and morphological characteristics of S100a11 lineage astrocytes.

To determine whether Olig2 knockout also altered molecular identity, we performed scRNA-seq on FAC-sorted GFP[+] cells at P7 from control and Olig2 knockout brains electroporated at E16.5 (Fig. 4a). Both datasets contained three major cell types: neuronal lineage cells, astrocytes, and oligodendrocytes/OPCs (Supplementary Fig. 12a, left). Knockout brains showed a modest increase in the overall number of

astrocytes (Supplementary Fig. 12a, right), consistent with our immunostaining (Fig. 4b, c), while neuronal numbers remained unchanged (Supplementary Fig. 12a, b). A slight reduction in oligodendrocytes/OPCs was observed (Supplementary Fig. 12c), aligning with previous studies showing that Olig2 is essential for oligodendrocyte development[35–40]. Classic in vitro studies suggested that some RGCs could generate both astrocytes and oligodendrocytes/OPCs[41,42]. Leveraging the clonal information, we identified that ~40% of clones were astrocyte-specific, 40% were mixed (containing both astrocytes and oligodendrocytes/OPCs), and 20% were oligodendrocytes/OPCs-specific (Supplementary Fig. 12d, e). Following Olig2 knockout, mixed and oligodendrocytes/OPCs-specific clones were significantly reduced, while astrocyte-only clones increased (Supplementary Fig. 12e). This shift suggests that bipotent and oligodendrocyte-restricted RGCs failed to generate OPCs after Olig2 loss, which could explain the slight decrease in oligodendrocytes/OPCs and corresponding rise in astrocytes.

To examine molecular changes in astrocytes, we isolated astrocytes from control and Olig2 knockout datasets (Fig. 4i) and performed differential gene expression analysis. As expected, Olig2 lineage genes were downregulated (Fig. 4j, Supplementary Fig. 12f, g and Supplementary Data 7), including those directly positively regulated by Olig2 (e.g., Serpine2, Nap1l5, Ednrb; Supplementary Fig. 10f, 12h). Conversely, S100a11 lineage markers were upregulated, among which were genes normally repressed by Olig2 (e.g., Aldoc, Trim2, Thrsp; Supplementary Figs. 10f, 12h). These findings indicate that Olig2 knockout reshapes the astrocyte molecular profile, downregulating Olig2 lineage genes while upregulating S100a11 lineage markers, suggesting a shift in lineage identity. To confirm this, we annotated the astrocytes into the five previously defined subtypes (Fig. 4i). Consistent with our immunostaining (Fig. 4b, c), the proportion of Olig2 lineage astrocytes decreased, while that of S100a11 lineage astrocytes increased (Supplementary Fig. 12i). Clonal analysis showed more clones shared between the two lineages after Olig2 knockout (Fig. 4k, Supplementary Fig. 12j), and lineage coupling analysis revealed a closer clonal relationship between lineages (Fig. 4l). These findings highlight Olig2 as a key regulator of astrocyte lineage identity; its loss shifts differentiation toward the S100a11 lineage, supporting a model in which distinct RGC lineages contribute to cortical astrocyte heterogeneity.

## Perturbation of Olig2 lineage is associated with altered synapse formation in cortical neurons

The molecular diversity and spatial organization of astrocyte subtypes (Fig. 1) suggest that they may fulfill specialized functions[4]. To explore this, we conducted gene ontology analysis to identify genes associated to key astrocyte functions, such as blood-brain barrier (BBB) regulation, neuron projection development, brain homeostasis, astrocyte activation, synapse formation, and metabolic support of neurons (Supplementary Data 8). Based on these gene sets, we computed expression scores for each astrocyte subtypes (Fig. 5a). This analysis suggested that the Ast_Slc7a10 mainly supports synapse formation and BBB regulation, Ast_Serpinf1 is involved in metabolic support, Ast_Sfrp1 in brain homeostasis, and Ast_S100a4 in neuron projection development and activation, implying that astrocytes molecular heterogeneity underlies functional specialization.

Focusing on synaptogenesis, we found that S100a11 lineage astrocytes express synapse-inhibitory genes (Sparc[43] and Pcdh8[44]), while Olig2 lineage astrocytes express synapse-promoting genes[45], including Sparcl1[43], Chrdl1[46] and Pcdh17[47] (Supplementary Fig. 13a). These patterns, combined with higher synapse formation score, point to Olig2 lineage astrocytes (especially the Ast_Slc7a10) as key promoters of synaptogenesis. To validate this, we used our Olig2 knockout model, which showed a reduced proportion of Olig2 lineage astrocytes, particularly Ast_Slc7a10 (Fig. 4c; Supplementary Fig. 12i). Synapse-formation scores were lower in Olig2 knockout astrocytes

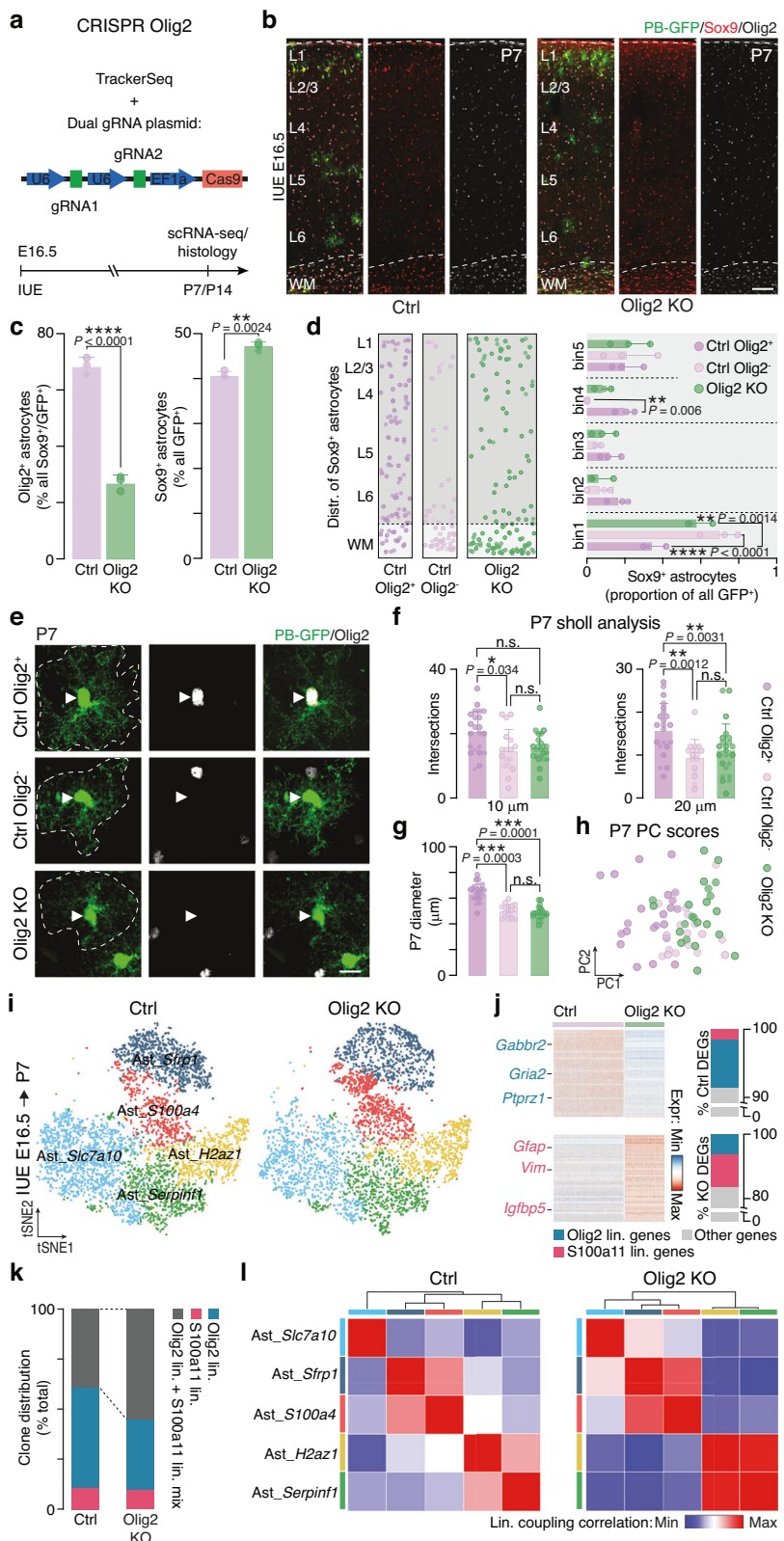

compared to control Olig2[+] and Olig2[−] astrocytes (Fig. 5b), supporting the idea that astrocytic contribution to synapse formation is compromised upon *Olig2* knockout. Moreover, *Olig2* knockout resulted in an increased number of Sparc[+] astrocytes (Supplementary Fig. 13b) and fewer Sparcl1[+] astrocytes (Supplementary Fig. 13c), both changes potentially contributing to disrupted synaptic development. However, scRNA-seq analysis at the subtype level showed only a modest increase

in *Sparc* and no significant change in *Sparcl1* expression in Ast_*Slc7a10* following *Olig2* knockout (Supplementary Fig. 13d). This suggests that the reduced synapse formation score may be primarily driven by shifts in astrocyte subtype proportions, with an increase in S100a11 lineage astrocytes (Sparc[+]), and a decrease in Ast_*Slc7a10* (Sparcl1[+]). To test functional outcomes, we measured spine density in neurons from the electroporated regions in both control and *Olig2* knockout brains at

**Fig. 4 | *Olig2* knockout impacts cortical astrocyte development. a** Experimental workflow of *Olig2* knockout and lineage tracing. **b** Control (Ctrl) and *Olig2* knockout (Olig2 KO) cortical columns stained for Sox9 and Olig2. Scale bar: 100 µm. **c**, Left: fraction of Olig2⁺ astrocytes in Ctrl and Olig2 KO cortices. Right: fraction of Sox9⁺ astrocytes among GFP⁺ electroporated cells. *n* = 3 animals; two-tailed *t*-test. **d** Distribution of Sox9⁺ astrocytes in the cortical column, comparing Ctrl Olig2⁺, Ctrl Olig2⁻, and Olig2 KO. Histograms show distribution across five cortical bins. *n* = 3 animals; two-way ANOVA with post hoc Tukey test. **e** Morphology of P7 astrocytes classified into Ctrl Olig2⁺, Ctrl Olig2⁻, and Olig2 KO. Olig2 expression indicated by arrowheads. Dashed lines indicate astrocyte boundaries. Scale bar: 10 µm. **f** Sholl analysis of individual astrocytes measuring intersections at 10 µm and 20 µm from cell center. *n* = 62 astrocytes, coming from 6 individual animals (3 for Ctrl and 3 for Olig2 KO); one-way ANOVA with post hoc Tukey test. **g** Quantification of the diameter of Ctrl Olig2⁺, Ctrl Olig2⁻ and Olig2 KO astrocytes.

*n* = 62 astrocytes coming from 6 individual animals (3 for Ctrl and 3 for Olig2 KO); one-way ANOVA with post hoc Tukey test. **h** PCA plot showing clustering of P7 astrocytes based on recorded morphological parameters. **i** tSNE representation of astrocyte subtypes from Ctrl and Olig2 KO conditions, traced from E16.5 and collected at P7. **j** Left: differentially expressed genes (DEGs) between Ctrl and Olig2 KO astrocytes (full gene list in Supplementary Data 7). Six selected genes are labeled. Right: proportion of lineage-specific genes (full gene list in Supplementary Data 4) shared with Ctrl or KO DEGs. **k** Proportion of clones unique to Olig2, unique to S100a11, and mixed lineage. **l** Heatmap of lineage coupling scores between astrocyte subtypes in Ctrl and Olig2 KO conditions. Values range from positive (red, coupled) to negative (blue, anti-coupled). n.s. not significant, *$p < 0.05$, **$p < 0.01$, ***$p < 0.001$, ****$p < 0.0001$. Values are shown as mean ± s.d.. IUE: *in utero* electroporation; WM: white matter; L: layer. Source data are provided as a Source Data file.

P21 and P50. Golgi staining revealed that while the overall neuronal structure remained intact (Fig. 5c), a closer examination of the apical dendrites showed a reduction in spine number at both ages (Fig. 5d, e), indicating potential impairment in synapse formation. We further confirmed this by labeling synapses using the presynaptic marker Vglut1 and the postsynaptic marker Psd95[48,49]. In control brains, synapse density was lower within the territory of electroporated Olig2⁻ astrocytes compared to Olig2⁺ astrocytes (Fig. 5f), suggesting that the latter plays a more active role in synapse formation. In *Olig2* knockout mice, there was a further reduction in synapse density (Fig. 5f), confirming that Olig2 lineage astrocytes are crucial for the proper formation of excitatory synapses. Similar disruptions were observed in inhibitory synapse density, as indicated by quantification of Gad2 and Gephyrin synaptic puncta (Supplementary Fig. 13e)[50,51]. While the observed reduction in oligodendrocytes/OPCs (Supplementary Fig. 12c) may also contribute[52–54], our data suggest that the synaptic defects are at least partly due to the loss of Olig2⁺ astrocytes. Together, these results support a model in which astrocyte subtypes have specialized functions, with Olig2 lineage astrocytes promoting synapse formation.

## Discussion

Our data provide a comprehensive perspective on how astrocyte heterogeneity arise during mouse neocortex development. We identified two distinct RGC subtypes: the classical *Emx1*⁺ RGC, which sequentially generates neurons and astrocytes[11,12], and a second RGC subtype that primarily gives rise to a specific subset of astrocyte with limited neuronal output. These RGCs coexist and together generate all dorsally derived cortical astrocytes (Fig. 5g). Clonal analysis revealed a substantial proportion of mixed clones containing astrocytes (from both lineages) and oligodendrocytes/OPCs (Supplementary Fig. 12e). This finding aligns with recent human[21,55] and mouse studies[56–58] identifying gliogenic tri-IPCs capable of producing astrocytes, oligodendrocytes/OPCs, and olfactory bulb interneurons, and supports a model in which both RGC_1 and RGC_2 generate astrocyte lineages through intermediate progenitor cells.

By combining *in utero* electroporation with the piggyBac transposase system, we achieved stable genomic labeling of cortical astrocytes derived from the dorsal pallium. This approach revealed five molecularly distinct astrocyte subtypes in both P7 and adult mouse neocortex, each occupying defined positions along the cortical column. The most prominent differences were observed among astrocytes in the GM, WM, and layer 1, consistent with previous studies[2,7,8]. Within the GM, we observed subtype-specific enrichment across upper, middle, and deep layers, supporting a layered astrocyte organization[2,7]. Using layer-specific gene sets from previous studies[2,7], we found that Ast_Slc7a10 primarily expressed upper- and middle-layer genes; Ast_Sfrp1, lower-layer and layer-1 genes; Ast_Serpinf1, genes spanning middle, lower, and layer 1; while Ast_H2az1 showed no clear layer preference (Supplementary Fig. 14). These profiles aligned

with spatial distributions obtained from MERFISH and Visium spatial transcriptomics (Fig. 1e, f, Supplementary Fig. 3g). We also identified a WM-specific subtype, Ast_S100a4, resembling a recently described S100a6⁺ astrocyte population in adult WM[17]. Altogether, these findings provide an expanded and integrative view of cortical astrocyte diversity, spanning the full depth of the cortical column—from layer 1 to the WM. This comprehensive sampling enabled the identification and spatial-molecular characterization of all dorsally derived astrocyte subtypes, further supporting the emerging view that astrocytes constitute a molecularly diverse population.

The precise contributions of intrinsic and extrinsic mechanisms to astrocyte heterogeneity remain unclear, and no consensus has yet emerged on how this diversity is established during cortical development. A recent fluorescence-based labeling study suggested that astrocyte colonization of the neocortex occurs randomly, rather than through clonally guided intrinsic programs[13]. In parallel, neuronal layers have been shown to influence astrocyte subtype formation and spatial organization, with lamination mutants like *Reeler* and *Satb2*-KO mice exhibiting disrupted astrocyte layering and subtype identity[7,8]. Conversely, another lineage-tracing study found that protoplasmic and pial astrocytes arise from distinct lineages, supporting intrinsic regulation of astrocyte diversity[59]. To directly address this, we employed TrackerSeq[10], an electroporation-based technique that labels RGCs independently of Cre drivers, to simultaneously uncover clonal relationships (*i.e.*, origins) and transcriptomic features (*i.e.*, molecular identities) at the single-cell resolution. This approach unveiled the critical contribution of clonality, *i.e.*, cell-intrinsic mechanisms, in driving the emergence of distinct cortical astrocyte subtypes. However, our results also suggest that extrinsic factors may also play a role in astrocyte fate specification. Although RGC_1 and RGC_2 are molecularly distinct and display biased lineage outputs, RGC_1 can give rise to both astrocyte lineages, indicating that cell-extrinsic mechanisms (*e.g.*, local environment) can modulate lineage output.

We identified two molecularly distinct RGC subtypes, each exhibiting unique proliferative behavior. One derives from the classical *Emx1*⁺ RGC (RGC_1), which produce both excitatory neurons and astrocytes. This findings aligns with previous MADM lineage tracing using *Emx1::CreER^{T2}* mice, where ~16% of neuronal clones also contain glial cells[11], and supports the notion that all neuronal subtypes arise from common *Emx1*⁺ RGCs[11]. Remarkably, we found that not all cortical astrocytes derived from this lineage. A substantial fraction originates from a distinct RGC population present from the onset of corticogenesis. This discrepancy with MADM studies likely reflects the limited labeling of weak or non-*Emx1*-expressing RGCs (RGC_2) by *Emx1::CreER^{T2}* mouse line[11]. Notably, we found that astrocytes from this second RGC pool emerge earlier during corticogenesis than those derived from the neuronal lineage. This is consistent with prior studies showing that embryonically generated astrocytes (between E16.5 and P0) preferentially occupy upper GM layers, while postnatally born

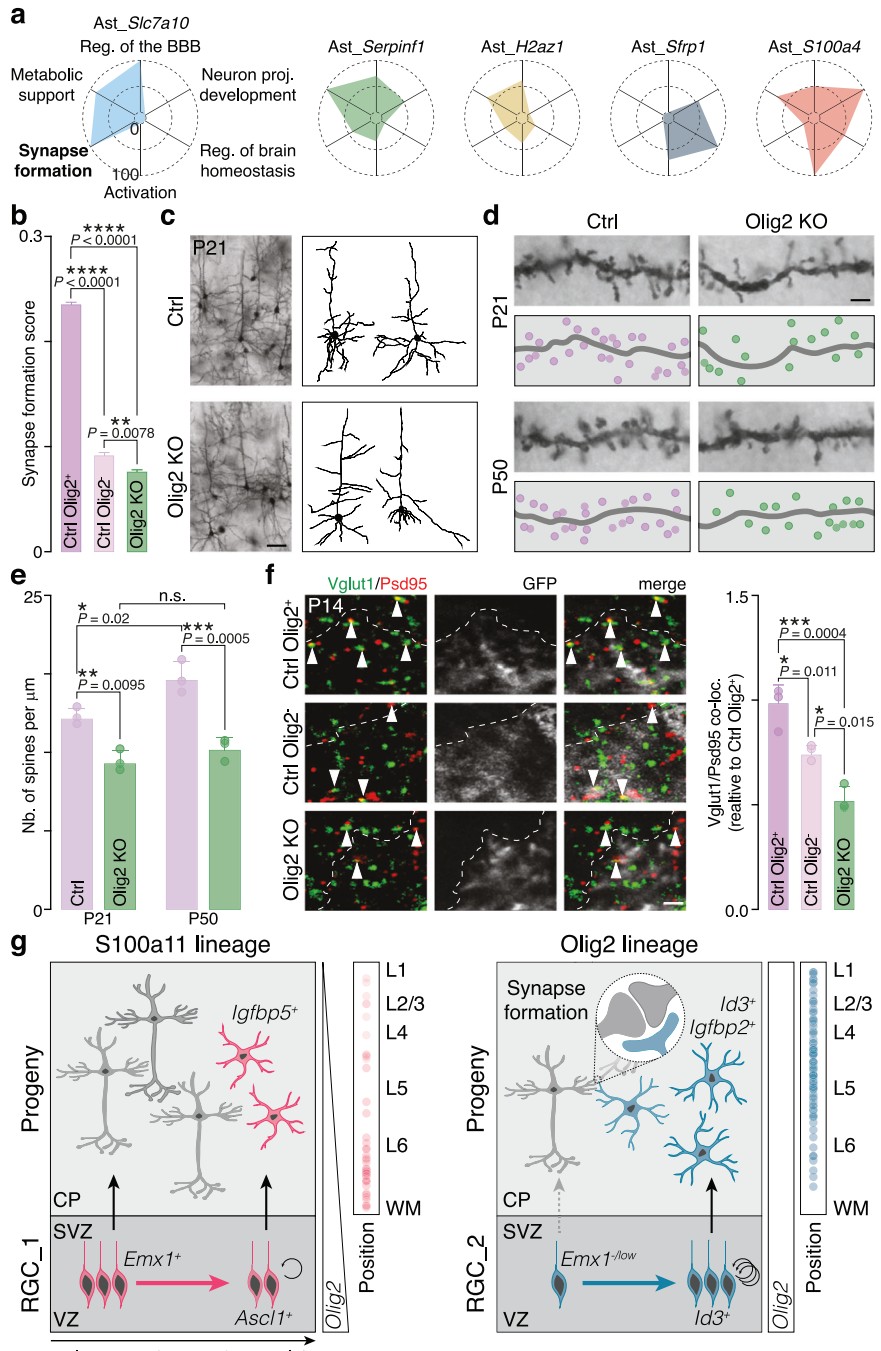

**Fig. 5 | Olig2 lineage astrocytes regulate synapse formation. a** Scaled expression scores for each gene set related to six primary astrocyte functions across various astrocyte subtypes (full gene list in Supplementary Data 8). **b** Synapse formation scores for control (Ctrl) Olig2⁺, Ctrl Olig2⁻, and *Olig2* knockout (KO) astrocytes. *n* = 6238 astrocytes (Ctrl Olig2⁺), 2360 astrocytes (Ctrl Olig2⁻), 4848 astrocytes (Olig2 KO); one-way ANOVA with post hoc Tukey test. **c** Left: P21 L2/3 cortical pyramidal neurons from Ctrl and Olig2 KO brains within the region of electroporated astrocytes. Golgi staining was used to evaluate neuronal morphology. Right: morphological reconstructions of neurons from Ctrl and Olig2 KO cortices. Scale bar: 25 μm. 6 neurons from 3 animals were analyzed from each condition. (**d, e**) Apical dendrites of neurons from Ctrl and Olig2 KO cortices at P21 (top) and P50 (bottom), illustrating spine abundance as quantified in (**e**). *n* = 3 animals; one-way ANOVA with post hoc Tukey test. Scale bar: 1.5 μm. **f** Left: excitatory synapses within the GFP⁺ astrocyte territory for Ctrl Olig2⁺, Ctrl Olig2⁻, and Olig2 KO astrocytes at P14. Scale bar: 2 μm. Dashed lines indicate astrocyte boundaries, and arrowheads highlight co-localized synaptic puncta. Right:

quantification of co-localized puncta in GFP⁺ astrocyte territories. *n* = 3 animals; one-way ANOVA with post hoc Tukey test. Scale bar: 2 μm. **g** Schematic representation of astrocyte lineage emergence in the mouse neocortex. Two distinct RGC subtypes (RGC_1 and RGC_2) give rise to two astrocyte lineages (S100a11 and Olig2). These RGC diverge over time, exhibiting distinct gene expression profiles and division rates. Both subtypes transiently express *Olig2* around E16.5, with *Olig2* expression maintained in the Olig2 lineage but downregulated in the S100a11 lineage. The predicted spatial distribution of these astrocyte lineages within the cortical column is shown on the right. Intermediate stages such as IPCs were deliberately omitted to preserve clarity and to reflect only the lineage relationships directly supported by the present study. n.s. not significant, *p < 0.05, **p < 0.01, ***p < 0.001, ****p < 0.0001. Values are shown as mean ± s.d. (**e, f**) or mean ± s.e.m. (**b**). Reg.: Regulation, proj.: projection, BBB: Blood-Brain Barrier; WM: white matter; VZ: ventricular zone; SVZ: subventricular zone; L: layer; CP: cortical plate. Source data are provided as a Source Data file.

astrocytes tend to locate in deeper layer[60,61]. The delayed production of astrocyte in S100a11 lineages likely reflects the fact that the *Emx1*+ RGCs must undergo the neurogenesis-to-gliogenesis transition before astrogenesis begins.

*Olig2* is a key transcription factor required for oligodendrocytes specification[62] and has also been implicated in astrocyte development[29–31]. Using CellOracle, we uncovered the gene regulatory network surrounding Olig2, identifying candidate upstream regulators and downstream targets that may underlie its role in lineage specification. Furthermore, we demonstrated that *Olig2* is essential for the specification and differentiation of Olig2 lineage astrocytes. *Olig2* knockout caused a shift in astrocyte identity towards the S100a11 lineage, reflected by both molecular and morphological changes, and a reduction in Olig2 lineage astrocytes. Consistent with previous observations[29], we observed an increase in GFAP expression, a marker enriched in S100a11 lineage. Although our knockout strategy does not selectively target RGC_2, and a potential role for Olig2 in RGC_1 cannot be fully excluded, our analyses center on its sustained, lineage-specific function in RGC_2 and mature astrocytes. Notably, Olig2 is naturally downregulated during the development of the S100a11 lineage, suggesting that Olig2 deletion is likely to have minimal impact in this population. Together, these findings highlight the importance of cell-intrinsic gene programs and clonal origin in shaping astrocyte diversity.

While astrocytes perform diverse functions, how these relate to molecular subtypes remains unclear. Our analysis suggests that distinct astrocyte subtypes carry out specialized roles in the neocortex. Notably, we show that Olig2 lineage astrocytes might play an important role in promoting synapse formation, consistent with recent reports of Olig2+ astrocytes localizing near excitatory synapses[30,63]. This function is supported by their higher expression of *Sparcl1*, a known positive regulator of synaptogenesis, whereas S100a11 lineage astrocytes express *Sparc*, which antagonizes the synaptogenic effects of *Sparcl1*[43]. *Olig2* knockout reduced the proportion of Olig2 lineage astrocytes and increased S100a11 lineage astrocytes, potentially shifting toward a less synaptogenic environment. Additionally, genes involved in synapse formation, such as members of the γ-protocadherins family[64,65], are preferentially expressed in Olig2 lineage astrocytes and downregulated upon loss of *Olig2*. While some effects may stem from reduced oligodendrocytes/OPCs, our findings support a model in which distinct astrocyte lineages differentially regulate synaptogenesis, with some subtypes promoting and others inhibiting synapse formation.

Our study provides evidence for two distinct astrocyte trajectories also in the human neocortex, suggesting that astrocytes lineage diversity is a mammalian evolutionary feature. In contrast, non-mammalian vertebrates, like reptiles and birds, predominantly possess Olig2 lineage astrocytes, lacking this complexity. The emergence of an RGC capable of generating both neurons and astrocytes likely contributed to this diversification. Notably, the appearance of this additional trajectory coincides with the neocortical evolution[66], indicating that astrocyte lineage diversification may have supported the increased molecular and functional complexity of the mammalian neocortex.

Taken together, our data demonstrate that cortical astrocyte subtypes exhibit distinct molecular, morphological, spatial and functional features and, importantly, originate from two RGC subtypes that coexist throughout corticogenesis. A deeper understanding of these lineage relationships holds the potential to uncover astrocyte subtype-specific mechanisms underlying astrocyte-related developmental disorders, including RASopathies and autism spectrum disorders[67–70]. In parallel, astrocytes have emerged over the past decade as promising candidates for targeted neuronal reprogramming[71,72]. Notably, subtypes such as S100a11 lineage astrocytes, which share a developmental origin with excitatory neurons, may be more amenable to reprogramming into neurons, an intriguing possibility that remains to be tested. This lineage-based susceptibility could provide a unique opportunity to improve and refine cell therapy approaches for brain repair.

## Methods

All experimental procedures described in this study were conducted in full compliance with Swiss legislation and were approved by the Geneva Cantonal Veterinary Authority (authorization no. 34938) and the Service de la Consommation et des Affaires Vétérinaires of the Canton of Vaud (authorization no. VD3755).

### Mouse strain

Female mice of the CD1 and C57BL/6 strains, obtained from Charles River Laboratory, were utilized in this study. The embryos were staged based on days post-coitus, with E0.5 defined as 12:00 on the day following the detection of a vaginal plug after overnight mating. For lineage tracing of *Emx1*+ RGCs, Emx1-IRES-Cre mice (B6.129S2-Emx1tm1(cre)Krj/J, Jackson number: 005628) were crossed with either Ai14 mice (B6.Cg-Gt(ROSA)26Sortm14(CAG-tdTomato)Hze/J, Jackson number: 007914) or ROSAnT-nG mice (B6;129S6-Gt(ROSA)26Sortm1(CAG-tdTomato*,-EGFP*)Ees/J, Jackson number: 023035). Brain tissue from these lines were kindly provided by Magdalena Götz laboratory. All mice were housed in the institutional animal facility, with a standard 12-h light and 12-h dark cycle and had access to food and water ad libitum. Both sexes were included in all the animal experiments unless stated otherwise.

### In utero electroporation

In utero electroporation was conducted as previously described[73]. Briefly, timed pregnant CD1 mice were anesthetized using isoflurane (5% for induction, followed by 2.5% during the surgical procedure), and they received the analgesic Temgesic from Reckitt Benckiser (10% in 0.9% NaCl). Embryos were injected in lateral ventricle with ~1 μl of DNA plasmid solution (diluted in endotoxin-free water and 0.002% Fast Green FCF (Sigma)). For the TrackerSeq experiments, the plasmid solution comprised 0.5 μg/μl of Helper plasmid and 0.5 μg/μl of Donor plasmid[10]. In the case of the Olig2 knockout experiments, the plasmid solution consisted of 0.5 μg/μl of Helper plasmid, 0.5 μg/μl of Donor plasmid, and 0.5 μg/μl of the Olig2 CRISPR plasmid (pX458-Ef1a-Cas9-H2B-mCherry, Addgene #171101). The electroporation of embryos was achieved by securing each head between circular tweezer-electrodes (5 mm in diameter, Sonidel), positioned across the uterine wall. Subsequently, a square-wave electroporator (Nepa Gene, Sonidel) delivered either 5 electric pulses at 25 V for 50 ms at 1 Hz for E12.5 embryos, or 5 electric pulses at 45 V for 50 ms at 1 Hz for E16.5 embryos.

### Immunohistochemistry, image acquisition, processing and quantification

For immunofluorescent staining, postnatal mice underwent cardiac perfusion with a 4% PFA solution, after which their brains were fixed by immersion in 4% PFA at 4 °C overnight, and subsequently stored in PBS at 4 °C. Coronal brain sections of 70 μm in thickness were generated using a vibrating microtome (Leica, VT1000S). In the case of embryonic brains, they were fixed for a duration of 4 h using 4% PFA, cryoprotected by immersion in 20% sucrose-PBS at 4 °C overnight. They were then embedded in O.C.T. (CellPath, KMA-0100-00A), snap-frozen at −70 °C in isopentane and stored at −80 °C. Coronal sections with a thickness of 16 μm were produced employing a cryostat (Leica, CM3050). For the staining process, both types of sections were initially subjected to a 1-h pre-incubation at room temperature within a solution that simultaneously blocked and permeabilized, consisting of 5% bovine serum albumin and 0.3% Triton X-100 in PBS. Subsequently, they were incubated with primary antibodies overnight at 4 °C. Afterward, the sections underwent three rinses in PBS and were then exposed to corresponding Alexa-conjugated secondary antibodies (1:1000 dilution; Donkey anti-Rabbit IgG, Alexa Fluor 488 Invitrogen

A11055; Donkey anti-Rabbit IgG, Alexa Fluor 647 Invitrogen A31573; Donkey anti-Goat IgG, Alexa Fluor 488 Invitrogen A21206; Donkey anti-Rabbit IgG, Alexa Fluor 405 Invitrogen A48258; Donkey anti-Chicken IgG, Alexa Fluor 488 Invitrogen A78948; Donkey anti-Rat IgG, Alexa Fluor 594 Invitrogen A21209; Donkey anti-Goat IgG, Alexa Fluor 405 Abcam ab175664) for 2 h at room temperature. Finally, the sections were mounted using Fluoroshield with DAPI (Sigma-Aldrich, F6057) or Fluoromount™ Aqueous Mounting Medium (Sigma-Aldrich, F4680). The primary antibodies used, along with their dilutions, were as follows: chicken anti-GFP (Invitrogen, A10262, 1:1000), rat anti-RFP (Chromotek, 5F8, 1:500), rabbit anti-Sox9 (Merck Millipore, AB5535, 1:500), rabbit anti-GFAP (Abcam, ab7260, 1:500), rabbit anti-Neurod2 (Abcam, ab104430, 1:500), rabbit anti-Pax6 (Covance, PRB-278P, 1:500), mouse anti-Pax6 (BD Biosciences, 561462, 1:500), rabbit anti-GABA (Sigma-Aldrich, A2052, 1:500), rabbit anti-S100b (Abcam, ab41548, 1:500), goat anti-Olig2 (R&D Systems, AF2418, 1:200), rabbit anti-Sox2 (Abcam, ab97959, 1:500), guineapig anti-VGLUT1 (Synaptic System, 135304, 1:1000), mouse anti-PSD95 (Millipore, MAB1596, 1:600), goat anti-EGFR (R&D Systems, FAB9577B, 1:1000), goat anti-Sparc (R&D Systems, AF942; 1:200), goat anti-sparcl1(R&D Systems, AF2836, 1:1000), mouse anti-Gad2 (Merck, MAB5406; 1:500), guinea-pig anti-Gephyrin (Synaptic Systems, 147 318; 1:200), goat anti-Sox10 (Santa Cruz, sc-17342, 1:500), rat anti-Sox2 (Thermo Fisher, 14-9811-82, 1:500), rabbit anti-Id3 (Cell Signaling Technology, #9837, 1:500), mouse anti-Ascl1 (BD Biosciences, 556604, 1:200), rat anti-Ki67 APC-conjugated (Thermo Fisher, 17-5698-82, 1:500). For Ki67 staining, we always use Donkey anti-Rat IgG, Alexa Fluor 647 Invitrogen A48272. Images were captured using a ZEISS LSM 800 Airyscan confocal microscope. The morphological analysis of astrocytes was performed by using original confocal Z-stack images in ImageJ software (version 2.14.0/1.54 f). Due to the clonal distribution of cortical astrocytes, it is often difficult to identify isolated cells, as reflected in Fig. 4e. To address this, we first outlined the territory of each astrocyte prior to measuring morphological parameters, using the midline between adjacent astrocytes as the boundary. Psd95-Vglut1 and Gad2-Gephyrin colocalization was calculated within the L2/3 using the ImageJ software (version 2.14.0/1.54 f). Puncta were considered co-localized if the distance between them was ≤ 0.5 μm. Number of colocalized puncta was obtained and normalized to control group.

For Golgi staining, whole brains were harvested whole from P21 and P50 mice and stained using the FD Rapid GolgiStain kit (FD NeuroTechnologies). Extracted brains were rinsed with double distilled water and then immersed in a 1:1 mixture of FD Solution A:B for 2 weeks at room temperature in the dark. Brains were then transferred to FD Solution C and kept in the dark at 4 °C for 72 h. Solution C was replaced after the first 24 h. Brains were sectioned coronally at a thickness of 180 μm using a vibratome (Leica VT 1200 S). Coronal sections were transferred to gelatin coated slides (FD NeuroTechnologies) onto small drops of FD Solution C. Sections were dehydrated in ethanol, cleared in xylene, and covered with Permount® mounting medium. Only spines in secondary apical dendrites from superficial layer pyramidal neurons in matched somatosensory cortex were used for analysis. Pyramidal neurons were photographed using a Zeiss Axio Imager A2 microscope equipped with a 10x (for morphology analysis) or 100x objective (for spine number quantification). ImageJ software (version 2.14.0/1.54 f) was used to measure dendrite length and manually count spine number.

All the quantitative analyses were performed on the somatosensory cortex (for the analyses performed at P7, P14, P21 and P50) or putative somatosensory regions (for the analyses at E12.5, E16.5). When quantifying the proportion of astrocytes and analyzing their morphology, the following criteria were applied to ensure that only astrocytes were included: (a) cells had to be positive for astrocyte markers, either Sox9 or S100b; (b) cells were considered astrocytes if they exhibited characteristic astrocytic morphology, such as a star-shaped soma with numerous fine, radiating processes (for protoplasmic astrocytes), elongated, less-branched processes with linear orientation (for fibrous astrocytes), or long, horizontal processes extending parallel to the pial surface (for layer 1 astrocytes). Oligodendrocytes were excluded based on their compact cell bodies and fewer, shorter, non-ramified processes. Only cells clearly meeting both marker expression and astrocyte-specific morphological criteria were included in our quantifications. For the quantification of GFP[+], RFP[+], and GFP[+]/RFP[+] cells in *Emx1::cre, ROSA^{nT-nG}* mice, all images were acquired using the same laser intensity. During image processing, laser intensity was adjusted as follows: (a) for GFP acquisition, the laser intensity was reduced until no GFP[+] cells were visible in structures below the cortex—such as the medial/lateral ganglionic eminence at E12.5 and E16.5, or the striatum and thalamus at P7—since cells in these regions are not derived from *Emx1[+]* RGCs; (b) for RFP acquisition, the laser intensity was adjusted to clearly visualize the nuclei of interneurons in the medial/lateral ganglionic eminence (E12.5 and E16.5), or in the cortex (P7). Cells GFP[+] alone or GFP[+]/RFP[+] were classified as derived from *Emx1[+]* RGCs (RGC_1), while all remaining cells (GFP[-]/RFP[+]) were considered to originate from RGC_2.

## TrackerSeq library preparation

TrackerSeq represents a piggyBac transposon-based library designed explicitly for seamless integration with the 10x single-cell transcriptomic platform. The creation of the TrackerSeq library followed a previously established procedure[10]. In summary, the piggyBac donor plasmid (PBCAG-eGFP, Addgene #40973) underwent several modifications. First, a Read2 partial primer sequence was inserted into the 3′ UTR of the eGFP sequence. Subsequently, through six distinct Gibson Assembly reactions (NEB, #E2611S)[74], the synthetic lineage barcode oligo mix was cloned between the Read2 partial primer and the poly(A) tail of eGFP. These modifications enable the retrieval of barcodes by the 10x single-cell transcriptomic platform. Our barcode library preparations exhibit a diversity of at least $10^5$ range, enabling us to recover approximately $10^4$ barcodes in each experiment.

## Sample collection for scRNA-seq

Brains collected from E18.5 embryos were dissected on ice with Leibowitz medium with 5% FBS whereas brains collected from P7 pups were exposed and kept in bubbled EBSS with 5% FBS on ice then transferred to Hibernate A medium with 10% FBS and B27 (1:50 dilution), while being observed under a dissecting scope to identify the positive regions. GFP[+] cortices were then dissociated using the Papain dissociation system following the recommended protocol from Worthington (#LK003150), and further processed with the gentleMACS Dissociator following the manufacturer's instructions. To isolate and collect GFP[+] cells, flow cytometry was employed with the BD FACSAria™ Fusion. Initially, cell suspensions were gated based on DAPI-negative and forward scatter, and from within this population, cells expressing GFP were collected in bulk for subsequent processing on the 10x Genomics Chromium platform. In all FACS experiments, non-electroporated brain tissue served as a negative control to exclude background fluorescence.

We performed three 10x reactions using cells from two animals electroporated at either E12.5 or E16.5, with cell collection at P7. Similarly, for the *Olig2* knockout scRNA-seq datasets, three 10x reactions were conducted with cells from one control and one knockout animal, both electroporated at E16.5 and collected at P7.

## Construction of scRNA-seq and TrackerSeq libraries

For experiments involving the 10x Genomics platform, the following materials were employed: Chromium Single Cell 3′ Library & Gel Bead Kit v3.1 (PN-1000121), Chromium Single Cell 3′ Chip Kit v3.1 (PN-1000127), and Dual Index Kit TT Set A (PN-1000215). These reagents were utilized in accordance with the manufacturer's provided

instructions. The lineage/TrackerSeq barcode library amplification process followed the previously outlined procedure[10], using the standard NEB protocol for Q5 Hot Start High-Fidelity 2X Master Mix (#M094S) in a 50-µl reaction, with 10 µl of cDNA as the template. To be specific, each PCR reaction contained the following components: 25 µl Q5 High-fidelity 2X Master Mix, 2.5 µl of a 10 µM P7_indexed reverse primer, 2.5 µl of a 10 µM i5_indexed forward primer, 10 µl of molecular-grade H20, and 10 µl of cDNA.

### Processing of sequencing reads

Transcriptome libraries and TrackerSeq barcode libraries were subjected to sequencing using the NovaSeq 6000 system from Illumina, conducted at the iGE3 genomic platform at the University of Geneva and MPIB Next Generation Sequencing core facility. The sequencing data in FASTQ files were subsequently aligned to a reference transcriptome (mm10-2.1.0) and collapsed into Unique Molecular Identifier (UMI) counts using the 10x Genomics Cell Ranger software, versions 6.0.1. In the case of TrackerSeq, preprocessing of the reads in the R2 FASTQ files involved trimming the sequences situated to the left and right of the lineage barcodes (BC). Lineage barcodes shorter than 37 base pairs were excluded from further analysis. Cell barcodes (Cell) were extracted from the corresponding Seurat dataset to create a whitelist of cell barcodes. These extracted cell barcodes and UMIs were incorporated into the read names of the lineage barcode FASTQ files. The resulting FASTQ files underwent processing to generate a sparse matrix in CSV format. In this matrix, the rows represented individual cells identified by their unique cell barcodes, while the columns represented lineage barcodes. Only Cell–UMI–BC triples supported by a minimum of 10 reads and Cell–BC pairs with at least 6 UMIs were considered for subsequent analyses. The assignment of CloneIDs to cell barcodes was accomplished by clustering the matrix using Jaccard similarity and average linkage, following the methodology demonstrated by Wagner and colleagues[10,75].

### Lineage coupling z-scores and correlations analysis

We conducted an analysis to determine shared clones, lineage coupling z-scores, and correlations for each pair of cell types or cell subtypes, following previously established methods[10,75]. Briefly, a TrackerSeq clone was deemed "shared" if it included a minimum of 2 individual cells assigned to each state. We tallied the total count of shared TrackerSeq clones for each pair of cell types or cell subtypes. This count was then compared to randomized data in which assignments of cell types or cell subtypes were randomly shuffled. We performed 10,000 random permutations to calculate a z-score for each original count of "shared" clones in relation to the expected distribution by random chance. The lineage coupling z-scores were then subjected to hierarchical clustering and visualized as a heatmap. Positive lineage coupling z-scores indicated pairs of cell types or cell subtypes that shared significantly more TrackerSeq clone barcode hits than what would be expected by chance, while negative lineage coupling z-scores suggested significantly less coupling than expected by chance. Furthermore, we computed correlation coefficients between the z-scores for each pair of cell types or cell subtypes and represented these lineage coupling correlations as a clustered heatmap. For a more detailed explanation of the calculation of lineage coupling z-scores and correlations, please refer to the original TrackerSeq study[10].

### Data integration, annotation and subtype identification

All scRNA-seq analysis were performed in R (version 4.2.1) following the Seurat workflow (version 4.3.0.1)[76] for cell filtering and data normalization. Each dataset was initially imported into R as a count matrix, which was then transformed into a Seurat object using the Create-SeuratObject() function with the following parameters: min.cells = 3 and min.features = 200. Cells harboring more than 10% mitochondrial genes were excluded from further analysis. To integrate the P7

datasets with those from Di Bella et al.[16] (Supplementary Fig. 1c), we employed Harmony (version 0.1.1)[77] using the RunHarmony() function with default parameters. For the identification of Seurat clusters, we constructed a shared-nearest neighbor graph using the FindNeighbors() function, based on Harmony embeddings (dimensions = 30). This graph served as input for the SLM algorithm, implemented through the FindClusters() function in Seurat (dimensions = 30, res = 0.5). Concurrently, cell type scores were computed using known brain cell type markers (Supplementary Data 1) via the CellCycleScoring() function. Clusters were manually annotated based on these cell type scores, gene expression, as well as using the online available databases for the mouse brain (http://mousebrain. org)[78]. Clusters that could not be definitively assigned to a specific cell type were labeled as "Undefined." For plotting the expression of genes in scRNA-seq dataset, in individual cell, counts of each gene were scaled by the total number of transcripts, multiplied by a factor of 10,000 to account for sequencing depth, and then log-transformed using the natural logarithm (log1p). This process makes expression values comparable across cells and reduces the effect of technical variation.

To identify astrocyte subtypes at P7 and distinguish them from more immature states, we initially isolated all astrocytes and RGCs from the integrated dataset (Supplementary Fig. 1c). We then conducted a re-clustering step using Seurat functions FindNeighbors() and FindClusters() with dimensionality set to 40 and a resolution of 1.3 (Fig. 2a). Astrocyte clusters were named based on top-regulated genes and their annotation was kept throughout all the datasets of this study (Fig. 1b, Fig. 4i).

To identify astrocyte subtypes in adult mouse cortex, we employed the FindTransferAnchors() and TransferData() functions to transfer labels from the P7 scRNA-seq dataset (Fig. 2a) to an integrated astrocyte dataset derived from Bocchi et al.,[17] and Endo et al.,[18].

To identify RGC subtype- and astrocyte-lineages at E18.5 (Fig. 3b), we began by manually annotating the E18.5 dataset into general cell types (e.g., excitatory neurons, astrocytes, RGCs) based on Seurat clusters and cell type scores (Supplementary Data 1). To facilitate the annotation of cell subtypes, including RGCs and astrocyte lineages, we generated a reference dataset sourced from Di Bella et al.[16], which encompassed RGCs and glia cells across E17.5 to P1 timepoints.

For predicting RGC subtypes before E16.5 (Fig. 3d, Supplementary Fig. 7d), we employed the FindTransferAnchors() and TransferData() functions to transfer labels from the E18.5 dataset to a RGC dataset derived from the Atlas of Di Bella et al. (Supplementary Fig. 7d)[16]. This RGC dataset was created by isolating RGCs from E10.5, E12.5, E14.5, and E16.5 time points (E10.5-E16.5 dataset).

To identify astrocyte subtypes within the *Olig2* knockout dataset, we integrated both the control and *Olig2* knockout datasets using the Harmony function RunHarmony() with default parameters. Subsequently, we subset the astrocyte population and conducted a re-clustering step using the Seurat functions FindNeighbors() and FindClusters(). The dimensionality was set to 30, and the resolution was set at 0.8. Given the substantial overlap between the control and *Olig2* knockout astrocytes, and the presence of subtype labels for the astrocytes from the control dataset (Fig. 1b), we assigned the same labels to astrocytes from the *Olig2* knockout dataset that fell within the same cluster (Fig. 4i).

### Differential gene expression analysis

To identify markers specific to astrocyte subtypes (Fig. 1c), we used the Seurat function FindAllMarkers() with the following parameters: min.pct = 0.25 and logfc.threshold = 0.25. For the identification of markers associated with RGC subtypes (RGC_1 and RGC_2, Fig. 3e), we used the function FindMarkers() with parameters set to min.pct = 0.2 and logfc.threshold = 0.2. To identify genes that were differentially expressed between the control and *Olig2* knockout datasets (Fig. 4j),

we utilized the Seurat function FindAllMarkers() with the following parameters: min.pct = 0.25, logfc.threshold = 0.25, and only.pos = T. Genes with an adjusted *p*-value greater than 0.05 were retained for subsequent analysis.

## Cell cycle analysis

To determine the proportion of RGCs that fall into G2/M phase, we applied tricycle package[79] to the E18.5 scRNA-seq dataset. By applying tricycle function Runtricycle(), each cell is assigned a cell cycle phase (i.e., S, G1 and G2/M phase). The proportion of cells fall into different phases was calculated.

## Spatial transcriptomics analysis

MERFISH experiment was performed following instructions from VIZGEN MERSCOPE Platform. For tissue processing and sectioning: brains were collected from E14.5 embryos, P7 pups and 3-month-old mouse, in cold HBSS. The obtained tissue was rapidly fresh-frozen in cold isopentane and preserved at −80 °C. We performed 10 μm coronal sections using a Leyca CM3050 cryostat and collected them on MERSCOPE slides. Briefly, sections were postfixed for 10 min in fresh 4% PFA and preserved for 24 h in 70% EtOH. This was followed by the 457 gene panel incubation o/n at 37 °C (Supplementary Data 3). After the hybridization, the section was embedded in a gel solution and cleared. Finally, the slide was placed in the MERSCOPE instrument to run the acquisition pipeline (https://vizgen.com/). Gene expression matrix and coordinates were exported from the MERSCOPE Visualizer for each ROI and assembled in a Seurat project for gene expression analysis. Briefly, we manually defined ROIs (each one corresponding to a single cell) using the MERSCOPE visualizer (https://vizgen.com/vizualizer-software/). We then exported the raw gene counts per ROI (for the 457 detected genes, Supplementary Data 3) and processed single-cell expression following the Seurat workflow (version 4.3.0.1): (1) gene count normalization to the total expression and log-transformation, (2) highly variable genes detection and principal component analysis, (3) graph-based clustering (with the 30 first principal components and a clustering resolution of 0.8), (4) tSNE calculation. For cell type annotation, we calculated cell type scores for both adult and E14.5 MERFISH datasets and manually annotated Seurat clusters into cell types. To predict the position of astrocytes in the P7 and adult neocortex, we used our P7 astrocyte scRNA-seq dataset (Fig. 1b) as a reference, and transferred the astrocyte subtype labels or astrocyte lineage labels to both P7 and adult MERFISH datasets, via the functions FindTransferAnchors() and TransferData() from Seurat. Same procedure was performed for transferring the astrocyte subtype labels to the adult Visium dataset. For the prediction of the position of RGC subtypes in E14.5 MERFISH dataset, same functions were used with the E10.5-E16.5 dataset (Supplementary Fig. 7d) used as reference.

## Inference of developmental trajectories of astrocytes

To reconstruct the transcriptional trajectories underlying the specification and differentiation processes of the astrocyte population, URD[9] was employed. The analysis began by subsetting all RGCs and astrocytes (Fig. 2a) from the integrated dataset (Supplementary Fig. 1c), serving as the initial dataset for URD simulation. First, we calculated a diffusion map using Destiny v2.14.044 implemented in the function calcDM() from URD with knn = 200. As root cells, we choose a subset of RGCs at E10.5. Cells were ordered in accordance with pseudotime by simulating diffusion from the root and subsequently calculating the distance of each cell from the root. For this, we used the function floodPseudotime() with parameters: *n* = 100 (number of simulations) and minimum.cells.flooded = 2, followed by the functions floodPseudotimeProcess() and pseudotimeDetermineLogistic(). We further used pseudotimeWeightTransitionMatrix() with default parameters to determine the slope and inflection point of the logistic function used to bias the transition probabilities. As tip cells, we assigned P7 astrocyte subtypes, based on which we simulated

random walks on the cell-cell graph from each tip to the root using connections in the biased transition matrix via the functions simulateRandomWalk() and processRandomWalks() from URD. Specifically, 12,500 random walks were performed for each tip. Finally, the URD tree was built using function buildTree() with following parameters: visit.threshold = 0, minimum.visits = 0, bins.per. pseudotime.window = 2, cells.per.pseudo time.bin = 300, divergence. method = "ks", p.thresh = 0.025, dendro. node.size = 150, min.cells.per.segment = 500 and min.pseudotime.per. segment = 0.01.

## Gene cascade analysis

To identify marker genes specific for trajectories, we used the function aucprTestAlongTree() in the URD package to walk backward from each tip along the trajectory, making pairwise comparisons between the cells in each segment and the cells from each of that segment's sibling and children (segments with equivalent or higher pseudotime values)[9,16], with following parameters: log.effect.size = 0.25, auc.factor = 1, max. auc.threshold = 1, frac.must.express = 0.1, frac.min.diff = 0.1, genes. use = genes.use, root = 10, only.return.global = False and must.beat. sibs = 0.5. We also used the funcion markersAUCPR() to calculate marker genes for each tip with following parameters: effect.size = 0.4 and auc.factor = 1.1. We than combined marker genes identified from functions aucprTestAlongTree() and markersAUCPR(). Marker genes for Ast_Slc7a10, Ast_Serpinf1 and Ast_H2az1 were grouped as Olig2 trajectory marker genes, while those for Ast_Sfrp1 and Ast_S100a4 were grouped as S100a11 trajectory marker genes. To determine the 'on and off' timing of expression of each gene, we used function geneSmoothFit() from URD, which takes a group of genes and cells, averages gene expression (parameters: moving.window = 5, cells.per.window = 25, pseudotime.per.window = 0.005), and then uses smoothing algorithms (spline fitting) to describe the expression of each gene[16]. Genes were then ordered and visualized according to their expression along the pseudotime via the function determine.timing() provided in URD package. The gene cascade heatmap was split into two (i.e., S100a11 and Olig2 trajectories) after the first branching point of the URD tree.

## PCA distance

PCA distance was calculated to measure the molecular changes of RGC subtypes across ages (Supplementary Fig. 7e). Briefly, for each RGC subtype (i.e., RGC_1 and RGC_2) at each timepoints (i.e., E10.5, E12.5, E14.5 and E16.5), the first 30 PCA dimensions were considered. The average PCA distance within for each RGC subtype between E10.5 and the others timepoints was calculated using the "dist()" function from the R stats package. As a way of normalization, each average PCA distance between E10.5 and other ages was divided by a scrambled distance, which was calculated by randomly splitting the whole RGC dataset into 2 groups of cells and calculating their average PCA distance.

## Gene regulatory network analysis

To perform CellOracle GRN analysis, the publicly available 10x genomics E18 (https://www.10xgenomics.com/datasets/fresh-embryonic-e-18-mouse-brain-5-k-1-standard-2-0-0) developing mouse cortex multiome dataset was analyzed using Signac (version 1.14.0). After normalization, scaling, dimensionality reduction, and clustering as described in the provided code, E18 Di Bella et al. [16] single cell RNA-seq reference dataset was used to transfer annotations using Seurat (version 5.1.0). After annotations, 'Cycling Glial cells' and 'Astrocytes' were subset and cicero (version 1.3.9) was used to generate peak co-accessibility scores through monocle3 (version 1.3.7) and SeuratWrapper (version 0.3.2). Our dataset was imported as an anndata object into scanpy (version 1.9.3) through Seurat to h5ad conversion by SeuratDisk (version 0.0.0.9021). Top 3000 differentially expressed genes in the anndata were filtered. UMAP, PAGA, and ForceAtlas2 graph construction was performed. To end this step, one random cell

in the Cycling glial cell population was selected as the root_cell to perform pseudotime calculation using CellOracle (version 0.10.15) and anndata object was saved with the calculated pseudotime.

Co-accessibility peaks generated from the E18 subset multiome dataset through cicero was used as input for the CellOracle GRN ATAC analysis. Peaks at the transcription start site (TSS) of genes were annotated along with the corresponding co-accessible peaks. Next, CellOracle wrapper for gimmemotifs (version 0.17.0) was used to generate a base GRN containing motifs in peaks close to 2 kb from all gene promoters and co-accessibile peaks. Base GRN containing 'motifs to peak coordinates' and the final 'TF to peak coordinates' dataframes were saved as.csv files. GRN construction using the CellOracle (version 0.10.15) package was performed through known TF to gene interactions from online databases (TFLink)[80]. For all the E18.5 RGCs (i.e., RGC_1 and RGC_2), a.csv GRN file was generated, where "source" corresponds to upstream regulator and "target" corresponds to genes either positively (with positive coef) or negatively (negative coef) regulated by a source gene (Supplementary Data 6).

### Statistics, reproducibility and text editing

Randomly selected mice were assigned to different experimental groups. No further randomization was applied during data collection. Investigators were blinded to group assignment during the experiments and data analysis. No statistical methods were used to predetermine sample size, but our sample sizes are similar to those reported in previous publications[10,17]. No animals or data points were excluded from the analyses. All statistical tests were performed with GraphPad Prism (version 10.0.1) Software. Statistical significance was defined as $*p < 0.05$, $**p < 0.01$, $***p < 0.001$ and $****p < 0.0001$. All biological replicates (n) are derived from at least three independent experiments. All column graphs are expressed as median ± SD. The normality of the distribution of data points was verified using Shapiro-Wilk test. The Brown–Forsythe test was used to access the equality of group variances.

We used ChatGPT (OpenAI) to assist in language editing and text refinement.

### Reporting summary

Further information on research design is available in the Nature Portfolio Reporting Summary linked to this article.

## Data availability

The scRNA-seq data generated in this study have been deposited in the Sequence Read Archive under accession code PRJNA1027603. The MERFISH data generated in this study have been deposited in Zenodo under accession code 15632740. The processed datasets can be browsed at http://www.bocchilab.ch/Zhou_et_al_2025. The scRNA-seq datasets from Di Bella et al., were obtained from Gene Expression Omnibus under accession code GSE153164. The human scRNA-seq datasets from Trevino et al., were obtained from Gene Expression Omnibus under accession code GSE162170. The reptile scRNA-seq datasets from Tosches et al., were obtained from Sequence Read Archive under accession code PRJNA408230. The bird scRNA-seq datasets from Zaremba et al., were obtained from heiDATA. The Visium and scRNA-seq datasets generated from Bocchi et al., were obtained from Sequence Read Archive under accession PRJNA1125165. The mouse scRNA-seq datasets from Endo et al., were obtained from Gene Expression Omnibus under accession code GSE198027. Source data are provided with this paper.

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

## Acknowledgments

We thank the iGE3 Genomics Platform, Bioimaging, and FACS Facility at the University of Geneva, the Human Cellular Neuroscience Platform at Campus Biotech, as well as the NGS Core Facility at the Max Planck Institute of Biochemistry for their invaluable support; A. Benoit for technical assistance; the entire Jabaudon laboratory for their thoughtful feedback on the manuscript and for their constructive contributions throughout the project. We also thank the Magdalena Götz laboratory for generously providing brain tissues for lineage tracing using the Emx1-Cre mouse line. The Bocchi laboratory and this project are supported by the Swiss National Science Foundation (Ambizione grant: PZ00P3_201995).

## Author contributions

R.B. conceived the project and designed experiments. J.Z. and R.B. performed experiments and analyzed data with help of I.V., S.R.P., A.J., I.C., M.P., P.B., D.J., C.M., J.Z., and R.B. wrote the manuscript with input from all authors.

## Competing interests

The authors declare that they have no conflict of interest.
