## [Transparent Peer Review file · Nature Communications]

Dual lineage origins contribute to neocortical astrocyte diversity

Corresponding Author: Dr Riccardo Bocchi

Version 0:

Reviewer comments:

Reviewer #2

(Remarks to the Author)

The rebuttal was responsive to reviewer comments. Significant revisions were made to Figures 3e–3h and Extended Figure 9. The authors identified and validated markers for Prog_1 and Prog_2 populations. Based on this, they were able to quantify the Olig2+ population within both the S100a11 and Olig2 lineages. This provides more solid evidence supporting lineage development following their trajectory analysis (questions R1C3 and R2C5).

I have the following suggestions for text revisions:

1. Reviewers' comments indicated several caveats to interpretation of the data and so the author should add in qualifiers to conclusions in results and discussion, or perhaps separate 'limitations' section.
2. Further, my suggested title for authors' consideration is: Evidence for dual lineage origins of neocortical astrocytes.
3. Regarding the Olig2 knock out, classic papers (Zhou and Andersen, 2002, Cell; Lu et al, 2002, Cell) showed that loss of Olig function results in absence of oligodendrocytes and increased astrocyte production because Olig represses multipotent precursors that give rise to astrocytes. Removing reference to 'bi-potent' glial progenitors (lines 310, 311) is therefore appropriate in context of Olig2 function.

Reviewer #4

(Remarks to the Author)

After carefully reviewing the manuscript and the authors' detailed response letter addressing all reviewers' comments, I would like to provide the following evaluation:

First, while the reviewers provided highly professional and insightful comments, the authors' responses appear to have not fully addressed all the raised concerns.

In addition, I would like to raise several general concerns regarding the manuscript:

1. In the paper, the authors' consistent use of the term "progenitors" (e.g., "Emx1+ progenitors" and "Olig2+ progenitors" in the Abstract) creates unnecessary ambiguity, as this broad classification fails to distinguish between the distinct neural precursor populations active during corticogenesis. Given that radial glia represent the primary cortical stem cells - characterized by their unique bipolar morphology, ventricular zone localization, and expression of definitive markers - and serve as the principal source of both neuronal and glial lineages (including astrocytes, oligodendrocytes, and olfactory bulb interneurons), I strongly recommend adopting this more precise terminology.
2. Numerous studies (n>10) have conclusively demonstrated that cortical radial glia sequentially generate intermediate progenitor cells (IPCs) following neurogenesis, which can differentiate into both cortical astrocytes and oligodendrocytes, as well as olfactory bulb interneurons. Recent evidence from PMID: 39779846 and PMID: 39999176 further supports this developmental trajectory, while a MADM-based lineage tracing study (PMID: 33730566) has unequivocally shown that single radial glial cells can produce both astrocytes and oligodendrocytes. Given this well-established understanding of IPC multipotency, the proposed lineage relationships depicted in Figure 5G appear inconsistent with current neurodevelopmental paradigms and therefore require clarification or experimental validation. I note that authors also cited

Cell 184, 5053-5069 e5023, which showed cortical glial IPCs might give rise to both cortical astrocytes and oligodendrocytes.

3. A critical study using Olig2-Cre knockin mice (PMID: 33606177) has demonstrated through comprehensive fate mapping that this genetic line labels the entire cortical astrocyte population. This finding directly challenges the authors' conclusion about Olig2+ progenitors generating only a specific astrocyte subset. To reconcile these discrepancies, the authors should either: (1) replicate these lineage tracing experiments using their experimental conditions, or (2) provide a detailed discussion addressing how their results can be interpreted in light of this published evidence. The absence of such validation or commentary significantly weakens the manuscript's claims regarding dual lineage origins of neocortical astrocytes.

4. Even during the earliest stages of cortical patterning and neurogenesis, cortical radial glia exhibit significant heterogeneity rather than existing as a uniform population. As demonstrated in the paper PMID: 18165280, these radial glia consistently display spatial and molecular gradients that influence their neurogenic potential and differentiation fates. This intrinsic heterogeneity is well-documented and should be carefully considered when interpreting glial lineage relationships.

5. The Emx1-Cre knockin line represents a well-established tool for conditional gene knockout studies, demonstrating highly efficient recombination in both cortical radial glia and their progeny. During the critical developmental window from E10 to E14, this genetic driver achieves near-complete labeling of dorsal cortical radial glia populations, as extensively documented in numerous studies.

Version 1:

Reviewer comments:

Reviewer #4

(Remarks to the Author)

Having thoroughly reviewed this manuscript on multiple occasions, I find the current arguments remain unconvincing.

The authors must address these critical questions:"

1. R4C5. The authors must provide definitive evidence that Emx1-Cre does not exhibit ubiquitous expression in all dorsal cortical radial glia. This could be achieved through:

Systematic analysis of Emx1-Cre; Pax6-flox-cko; or Emx1-Cre; Sox2 flox-cko, or Emx1-Cre; Hopx flox-cko (conditional knockout models), demonstrating the persistence of: Pax6-, or, Sox2-, or Hopx-expressing cortical radial glial populations in the dorsal cortex of Emx1-cko mice.

The current reliance solely on a single Emx1-Cre reporter line fails to establish conclusive evidence regarding Emx1-Cre expression in the entire dorsal cortical radial glial population.

I must acknowledge having overlooked two critical issues during my initial review, for which I sincerely apologize. These substantive concerns include:"

2. The authors propose a highly unconventional model in their discussion and conclusion Fig. 5g, positing the existence of two distinct radial glial populations:

RGC_1: Generates both cortical excitatory neurons (ENs) and glia

RGC_2: Produces glia but no cortical excitatory neurons (ENs)

This claim raises fundamental biological questions that remain unresolved: during active neurogenesis, what is the evolutionary advantage or developmental purpose of maintaining a non-neurogenic radial glial population (RGC_2) in the cortex? Again, this study provides no strong direct lineage-tracing evidence for such a binary fate restriction among radial glia. The broader neuroscience community is likely to find this claim difficult to reconcile with established principles of cortical development.

3. The authors' interpretation in Extended Data Fig. 4 directly contradicts established findings from Trevino et al. (Cell, 2021; PMID: 34390642), which showed that S100A11 and KLF2 are expressed mainly in human cortical ependymal cells rather than astrocytes. To resolve this discrepancy and validate their claims, the authors must:

Provide feature plots of multiple canonical ependymal markers, including: FOXJ1, CRYAB, CXCL12, CCN1, CCN2 and S100A11, in Extended Data Fig. 4d and Fig 4e, f.

Given the availability of these standard markers and the simplicity of such validation experiments (which could be completed within a few minutes), this evidence is essential to support the study's conclusions."

Version 2:

Reviewer comments:

Reviewer #4

(Remarks to the Author)

Prior to reviewing the revised manuscript, I would like to provide feedback regarding the authors' responses to my previous comments.

R4C1. R4C5. The authors must provide definitive evidence that Emx1-Cre does not exhibit ubiquitous expression in all dorsal cortical radial glia.

Despite being emphasized twice in previous reviews, this critical issue has not been properly addressed in the authors' response.

In the seminal Emx1-Cre paper by Gorski et al. (2002) titled "Cortical excitatory neurons and glia, but not GABAergic neurons, are produced in the Emx1-expressing lineage" (J Neurosci. 22(15): 6309-14, PMID: 12151506), the authors clearly demonstrated that Emx1-Cre labels all pyramidal neurons and astrocytes in the dorsal cortex. This paper has been widely cited over 1,500 times. This finding, supported by the supplemental figure (attached) depicting recombination in the adult Emx1-cre; Z/AP brain. Specifically, panel (N) of the supplemental figure shows an X-gal-stained 40 µm coronal section of an adult Emx1-cre; Z/AP brain, revealing extensive recombination throughout the cortex and hippocampus. Notably, X-gal-positive staining was observed in pyramidal neurons and astrocytes.

[Redacted]

R4C2. The authors propose a highly unconventional model in their discussion and conclusion Fig. 5g, positing the existence of two distinct radial glial populations:

The author now claimed that in the abstract: The first lineage derives from Emx1+ radial glial cells that initially generate neurons and later switch to astrocyte production. The second lineage (derives from Emx1-low radial glial cells), with minimal neuronal output, predominantly produces a distinct subset of astrocytes marked by Olig2.

Given that nearly all cortical pyramidal neurons and astrocytes originate from Emx1-Cre-expressing radial glia (as established in prior literatures and my earlier comments), the authors' conclusion is untenable. Without strong direct evidence demonstrating the existence of Emx1-negative radial glia contributing to cortical astrocytes, this claim remains unsupported.

R4C3. The authors' interpretation in Extended Data Fig. 4 directly contradicts established findings from Trevino et al. (Cell, 2021; PMID: 34390642), which showed that S100A11 and KLF2 are expressed mainly in human cortical ependymal cells rather than astrocytes.

In their response, the authors once again failed to address my fundamental question. Specifically, the newly revised Extended Data Fig. 4 does not demonstrate the presence of S100a11-expressing astrocytes in the human cortex.

Furthermore, I note significant issues with the manuscript's scholarly rigor. On page 13, line 412 of the main text, key references supporting their argument have been omitted in this revision, despite being present in previous versions.

Taken together, these critical flaws lead me to conclude that the paper's major conclusions are fundamentally unsupported. The cumulative evidence suggests these findings cannot be accepted as scientifically valid.

Zhou et al, manuscript

Point-by-point answer to the Reviewers

Please note that all changes in the manuscript are now highlighted in **yellow**, while those specifically addressing the caveats to data interpretation are marked in **red**.

Reviewer #2:

The rebuttal was responsive to reviewer comments. Significant revisions were made to Figures 3e–3h and Extended Figure 9. The authors identified and validated markers for Prog_1 and Prog_2 populations. Based on this, they were able to quantify the Olig2+ population within both the S100a11 and Olig2 lineages. This provides more solid evidence supporting lineage development following their trajectory analysis (questions R1C3 and R2C5). I have the following suggestions for text revisions:

R2C1. Reviewers' comments indicated several caveats to interpretation of the data and so the author should add in qualifiers to conclusions in results and discussion, or perhaps separate 'limitations' section.

Response R2C1. We thank the reviewer for this helpful suggestion. We agree that there are caveats to the interpretation of our findings. While these were addressed throughout the Results section, they were not sufficiently discussed in the Discussion or clearly reflected in the conclusions. We have now added several statements in the Discussion to explicitly acknowledge these limitations and appropriately qualify our conclusions. These caveats are now highlighted in **red** in the manuscript.

R2C2. Further, my suggested title for authors' consideration is: Evidence for dual lineage origins of neocortical astrocytes.

Response R2C2. Thank you for the suggestion. We agree that the title should clearly reflect the main conclusion while remaining appropriately cautious. We propose the following revised title: "Dual lineage origins contribute to neocortical astrocyte diversity." This version maintains the core message while softening the claim, aligning with our preference to present the dual origin as a contributing factor rather than definitive evidence.

R2C3. Regarding the Olig2 knock out, classic papers (Zhou and Andersen, 2002, Cell; Lu et al, 2002, Cell) showed that loss of Olig function results in absence of oligodendrocytes and increased astrocyte production because Olig represses multipotent precursors that give rise to astrocytes. Removing reference to 'bi-potent' glial progenitors (lines 310, 311) is therefore appropriate in context of Olig2 function.

Response R2C3. We thank the reviewer for the meticulous inspection. As suggested, we have removed the term "bi-potent" to avoid potential misinterpretation regarding Olig2 function and revised the text accordingly for clarity (lines 307-308). Additionally, to provide broader context, we now cite the two aforementioned studies in relation to the observed reduction in oligodendrocyte/OPC populations and the concurrent increase in astrocyte numbers following Olig2 knockout (lines 307-308).

Reviewer #4:

After carefully reviewing the manuscript and the authors' detailed response letter addressing all reviewers' comments, I would like to provide the following evaluation:

First, while the reviewers provided highly professional and insightful comments, the authors' responses appear to have not fully addressed all the raised concerns.

In addition, I would like to raise several general concerns regarding the manuscript:

R4C1. In the paper, the authors' consistent use of the term "progenitors" (e.g., "Emx1+ progenitors" and "Olig2+ progenitors" in the Abstract) creates unnecessary ambiguity, as this broad classification fails to distinguish between the distinct neural precursor populations active during corticogenesis. Given that radial glia represent the primary cortical stem cells - characterized by their unique bipolar morphology, ventricular zone localization, and expression of definitive markers - and serve as the principal source of both neuronal and glial lineages (including

astrocytes, oligodendrocytes, and olfactory bulb interneurons), I strongly recommend adopting this more precise terminology.

Response R4C1. We thank the reviewer for this helpful suggestion. To improve clarity and precision, we have replaced the generic term "progenitor" with "radial glial cell" (RGC) throughout the manuscript where appropriate.

R4C2. Numerous studies (n>10) have conclusively demonstrated that cortical apical progenitor sequentially generate intermediate progenitor cells (IPCs) following neurogenesis, which can differentiate into both cortical astrocytes and oligodendrocytes, as well as olfactory bulb interneurons. Recent evidence from PMID: 39779846 and PMID: 39999176 further supports this developmental trajectory, while a MADM-based lineage tracing study (PMID: 33730566) has unequivocally shown that single apical progenitors can produce both astrocytes and oligodendrocytes. Given this well-established understanding of IPC multipotency, the proposed lineage relationships depicted in Figure 5G appear inconsistent with current neurodevelopmental paradigms and therefore require clarification or experimental validation. I note that authors also cited Cell 184, 5053-5069 e5023, which showed cortical glial IPCs might give rise to both cortical astrocytes and oligodendrocytes.

Response R4C2. We thank the reviewer for raising this important point. However, it is important to note that in our dataset we did not recover olfactory bulb interneurons and captured only a subset of oligodendrocytes/OPCs, as our collection time point (P7) precedes the end of oligodendrogenesis. Consequently, our dataset is not the most suited for a comprehensive assessment of lineage relationships among astrocytes, oligodendrocytes/OPCs, and olfactory bulb interneurons.

Nonetheless, our findings do not contradict the existence of tri-potent IPCs, instead, they are consistent with this model. In our TrackerSeq experiment, where radial glial cells were labeled at E16.5 and all progeny were collected at P7, we identified three clonal configurations: clones composed exclusively of astrocytes, clones composed exclusively of oligodendrocytes, and mixed clones containing both cell types (Extended Data Fig. 11e). Notably, approximately half of the clones were mixed, supporting the existence of IPCs capable of generating both astrocytes and oligodendrocytes. This TrackerSeq-based lineage tracing analysis aligns with the MADM study¹, which reported that ~25% of Emx1-derived clones included both astrocytes and oligodendrocytes. Quantifications of mixed clones containing astrocytes and oligodendrocytes/OPCs in the two astrocyte lineages (see graph below) revealed a similar proportion of mixed clones, supporting the idea that both lineages may harbor IPCs capable of producing astrocytes and oligodendrocytes. We have now cited related studies²⁻⁵ and acknowledged this tri-potent IPCs model in the revised manuscript (lines 411-416).

Regarding Figure 5g, the schematic was designed to highlight the molecular distinctions between radial glial cell subtypes and astrocyte lineages, and to integrate their functional specializations. It was not intended to depict the complete gliogenic hierarchy. Intermediate stages such as IPCs were deliberately omitted to preserve clarity and to reflect only the lineage relationships directly supported by our data. We now clarify this explicitly in the figure legend of Figure 5.

R4C3. A critical study using Olig2-Cre knockin mice (PMID: 33606177) has demonstrated through comprehensive fate mapping that this genetic line labels the entire cortical astrocyte population. This finding directly challenges the authors' conclusion about Olig2+ progenitors generating only a specific astrocyte subset. To reconcile these discrepancies, the authors should either: (1) replicate these lineage tracing experiments using their experimental conditions, or (2) provide a detailed discussion addressing how their results can be interpreted in light of this published evidence. The absence of such validation or commentary significantly weakens the manuscript's claims regarding dual lineage origins of neocortical astrocytes.

Response R4C3. We thank the reviewer for this important comment and apologize if the developmental dynamics of Olig2 expression across progenitor subtypes were not sufficiently clear in the manuscript. This point was also raised by Reviewer 1 and 2 during the last rounds of revision (see R1C3, R2C1) and was explicitly addressed in the revised text (see paragraph below, lines 262-272):

“Analysis of the E18.5 scRNA-seq dataset revealed that Olig2 is initially expressed in both progenitor subtypes but is rapidly downregulated as RGC_1 differentiates into S100a11 lineage astrocytes (Extended Data Fig. 9a, b, left). In contrast, its expression persists in Olig2 lineage astrocytes through the second postnatal week (Extended Data Fig. 9a, b, right)⁶. To validate this, we quantified Olig2⁺ cells across progenitors and astrocyte lineages using Pax6, Ascl1, and Id3 as markers. Since Pax6 labels both progenitors and astrocytes⁷, we classified Pax6⁺ cells in the cortical plate (CP) as astrocytes and those below as progenitors. Within progenitors, Ascl1 marked RGC_1, and Id3 identified RGC_2. In the CP, Id3⁺/Pax6⁺ cells represent Olig2 lineage astrocytes, and Id3⁻/Pax6⁺ cells the S100a11 lineage astrocytes (Extended Data Fig. 9c, d). Quantification revealed that ~60% of RGC_1 initially expressed Olig2, dropping to <20% in S100a11 lineage astrocytes. In contrast, >50% of RGC_2 were Olig2⁺, increasing to 80% in Olig2 lineage astrocytes (Extended Data Fig. 9e), supporting the expression pattern observed in the E18.5 scRNA-seq dataset.”

This biphasic expression explains why the Olig2::Cre knock-in line used by Li et al.² labels nearly all cortical astrocytes—Olig2 is indeed expressed in both RGC subtypes during early gliogenesis. However, our study shows that only RGC_2 maintains Olig2 expression and gives rise to a molecularly and functionally distinct lineage of astrocytes. In addition, this expression pattern of Olig2 from RGCs to astrocyte lineages reconcile with the observation that not all the cortical astrocytes are Olig2⁺ at P7. Therefore, rather than contradicting previous findings, our results offer a mechanistic framework to interpret them, providing new insight into the lineage-specific and temporally dynamic role of Olig2 during neocortical astrocyte development.

R4C4. Even during the earliest stages of cortical patterning and neurogenesis, cortical radial glia exhibit significant heterogeneity rather than existing as a uniform population. As demonstrated in the paper PMID: 18165280, these radial glia consistently display spatial and molecular gradients that influence their neurogenic potential and differentiation fates. This intrinsic heterogeneity is well-documented and should be carefully considered when interpreting glial lineage relationships.

Response R4C4. We fully agree with the reviewer that cortical radial glia is not a homogeneous population, and that spatial and molecular heterogeneity is present early during development, including along both the medio-lateral and rostro-caudal axes. As the reviewer rightly points out, these gradients in radial glial identity are likely to influence the fate potential and differentiation trajectories of their progeny, including astrocytes.

In the present study, however, we intentionally focused our analyses on the somatosensory cortex. By using *in utero* electroporation, we were able to target this region with spatial precision (see figure below showing a representative E16.5 electroporation in a P7 coronal section). This regional specificity is explicitly stated in the Materials and Methods section (lines 747-748, “Immunohistochemistry, image acquisition, processing, quantification and statistical analysis”).

We fully acknowledge the broader importance of radial glial heterogeneity and its potential impact on astrocyte diversity. This is a central question we are actively pursuing in a separate study focused on inter-regional variation in astrocyte subtypes. Preliminary analysis of MERFISH datasets from coronal sections at different rostro-caudal levels suggests that the five astrocyte subtypes identified in this study are broadly conserved across the cortex, though their relative abundance varies by region. These ongoing efforts aim to further elucidate how early spatial patterning of radial glia contributes to the regional specification of astrocyte identity.

R4C5. The *Emx1*-Cre knockin line represents a well-established tool for conditional gene knockout studies, demonstrating highly efficient recombination in both cortical apical progenitor and their progeny. During the critical developmental window from E10 to E14, this genetic driver achieves near-complete labeling of dorsal cortical apical progenitor populations, as extensively documented in numerous studies.

Response R4C5. We fully acknowledge that the *Emx1::Cre* line is a widely used and highly efficient tool for targeting dorsal cortical apical progenitors and has been instrumental in numerous conditional knockout studies. However, despite its extensive use, definitive quantification of the proportion of cortical astrocytes derived specifically from *Emx1*⁺ apical progenitors has not been established.

In our study, we directly addressed this question by employing a dual nuclear reporter strategy—crossing the *Emx1::Cre* line with the *ROSA^{nT-nG}* reporter—to differentially label *Emx1*-expressing progenitor-derived cells (GFP⁺) and non-*Emx1*-derived cells (RFP⁺). This approach enabled precise lineage discrimination at single-cell resolution. As expected, we confirmed the high recombination efficiency and specificity of the *Emx1::Cre* driver: nearly all Neurod2⁺ excitatory neurons were GFP⁺ and GABA⁺ inhibitory neurons were RFP⁺ (Extended Data Fig. 8a, b), fully consistent with prior literature.

Critically, this strategy also revealed a distinct population of *Emx1*⁻ progenitors (GFP⁻/RFP⁺) at E12.5 and E16.5, as identified using two independent progenitor markers—*Sox2* and *Pax6* (Extended Data Fig. 8c–f). Moreover, we observed a substantial fraction of cortical astrocytes at P7 that did not derive from *Emx1*⁺ progenitors (GFP⁻/RFP⁺), labeled with two independent astrocytic markers, *Sox9* and *S100b* (Fig. 3m, n; Extended Data Fig. 8g, h). To our knowledge, this is the first study to combine the *Emx1::Cre* driver with a double nuclear reporter system to systematically trace astrocyte lineage origin with this level of resolution. Similar results were observed using the *Ai14* non-nuclear reporter mouse line (Extended Data Fig. 8i), further validating the robustness of our observations. Collectively, these findings clearly demonstrate that not all cortical astrocytes originate from *Emx1*⁺ progenitors. Thus, while the *Emx1::Cre* line efficiently labels the majority of dorsal cortical progenitors, our data uncover the existence of an additional, previously unrecognized astrocyte-producing lineage that is not captured by *Emx1* expression.

References:

- 1 Shen, Z. *et al.* Distinct progenitor behavior underlying neocortical gliogenesis related to tumorigenesis. *Cell Reports* **34**, doi:10.1016/j.celrep.2021.108853 (2021).
- 2 Li, X. *et al.* Decoding Cortical Glial Cell Development. *Neuroscience Bulletin* **37**, 440-460, doi:10.1007/s12264-021-00640-9 (2021).
- 3 Gao, Y. *et al.* NOTCH, ERK, and SHH signaling respectively control the fate determination of cortical glia and olfactory bulb interneurons. *Proc Natl Acad Sci U S A* **122**, e2416757122, doi:10.1073/pnas.2416757122 (2025).
- 4 Wang, L. *et al.* Molecular and cellular dynamics of the developing human neocortex. *Nature*, doi:10.1038/s41586-024-08351-7 (2025).
- 5 Trevino, A. E. *et al.* Chromatin and gene-regulatory dynamics of the developing human cerebral cortex at single-cell resolution. *Cell* **184**, 5053-5069 e5023, doi:10.1016/j.cell.2021.07.039 (2021).
- 6 Cai, J. *et al.* A crucial role for Olig2 in white matter astrocyte development. *Development* **134**, 1887-1899, doi:10.1242/dev.02847 (2007).
- 7 Sakurai, K. & Osumi, N. The neurogenesis-controlling factor, Pax6, inhibits proliferation and promotes maturation in murine astrocytes. *J Neurosci* **28**, 4604-4612, doi:10.1523/JNEUROSCI.5074-07.2008 (2008).
- 8 Bellion, A., Wassef, M. & Metin, C. Early differences in axonal outgrowth, cell migration and GABAergic differentiation properties between the dorsal and lateral cortex. *Cereb Cortex* **13**, 203-214, doi:10.1093/cercor/13.2.203 (2003).

Zhou et al, manuscript

Point-by-point answer to the Reviewers

Reviewer #4:

Having thoroughly reviewed this manuscript on multiple occasions; I find the current arguments remain unconvincing. The authors must address these critical questions:

R4C1. R4C5. The authors must provide definitive evidence that Emx1-Cre does not exhibit ubiquitous expression in all dorsal cortical radial glia. This could be achieved through:

Systematic analysis of Emx1-Cre; Pax6-flox-cko; or Emx1-Cre; Sox2 flox-cko, or Emx1-Cre; Hopx flox-cko (conditional knockout models), demonstrating the persistence of: Pax6-, or, Sox2-, or Hopx-expressing cortical radial glial populations in the dorsal cortex of Emx1-cko mice.

The current reliance solely on a single Emx1-Cre reporter line fails to establish conclusive evidence regarding Emx1-Cre expression in the entire dorsal cortical radial glial population.

Response R4C1. We appreciate the reviewer's suggestion to use conditional knockout models (*e.g.*, Emx1-Cre;Pax6 flox, Sox2 flox, or Hopx flox) to assess Emx1-Cre specificity in dorsal cortical radial glia. However, we believe that our current approach provides an equivalent conceptual framework. These models are designed to delete the gene of interest in Emx1-expressing cells; thus, using immunostaining for the deleted gene to infer the presence of Emx1-negative progenitors is conceptually equivalent to the approach we already employed (see figure below). In particular, the dual reporter¹ allows Cre-dependent deletion of tdTomato, such that tdTomato expression is maintained exclusively in Emx1-negative cells. Therefore, tdTomato-positive cells can be interpreted in the same way one might interpret residual Pax6, Sox2, or Hopx expression in a conditional knockout context.

[Redacted]

[Redacted]

I must acknowledge having overlooked two critical issues during my initial review, for which I sincerely apologize. These substantive concerns include:

R4C2. The authors propose a highly unconventional model in their discussion and conclusion Fig. 5g, positing the existence of two distinct radial glial populations:

RGC_1: Generates both cortical excitatory neurons (ENs) and glia

RGC_2: Produces glia but no cortical excitatory neurons (ENs)

This claim raises fundamental biological questions that remain unresolved: during active neurogenesis, what is the evolutionary advantage or developmental purpose of maintaining a non-neurogenic radial glial population (RGC_2) in the cortex? Again, this study provides no strong direct lineage-tracing evidence for such a binary fate restriction among radial glia. The broader neuroscience community is likely to find this claim difficult to reconcile with established principles of cortical development.

Response R4C2. We appreciate the reviewer's concern but would like to clarify that the current version of the manuscript no longer presents a binary model of fate restriction between the two radial glial populations. This more rigid interpretation was indeed proposed in our original bioRxiv preprint. However, through multiple rounds of revision and the inclusion of additional analyses, we have substantially refined our interpretation. Our current conclusion is that RGC_1 predominantly gives rise to both neurons and astrocytes, while RGC_2 primarily generates astrocytes, though some RGC_2 cells can also produce neurons (Fig. 2k, 3c). We describe this not as a strict fate restriction but rather as a lineage bias or preference. This more nuanced view is explicitly illustrated in Figure 5g, where a dashed line from RGC_2 to neurons indicates their neurogenic potential. This interpretation is also reflected throughout the manuscript, in Abstract, Introduction, Results and Discussion sections (see text highlighted in red), and we have now ensured that it is clearly conveyed to the reader to avoid any possible misinterpretation.

R4C3. The authors' interpretation in Extended Data Fig. 4 directly contradicts established findings from Trevino et al. (Cell, 2021; PMID: 34390642), which showed that S100A11 and KLF2 are expressed mainly in human cortical ependymal cells rather than astrocytes. To resolve this discrepancy and validate their claims, the authors must:

Provide feature plots of multiple canonical ependymal markers, including: FOXJ1, CRYAB, CXCL12, CCN1, CCN2 and S100A11, in Extended Data Fig. 4d and Fig 4e, f.

Given the availability of these standard markers and the simplicity of such validation experiments (which could be completed within a few minutes), this evidence is essential to support the study's conclusions.

Response R4C3. We thank the reviewer for raising this important point. To specifically exclude potential contamination from ependymal cells, we utilized a list of ependymal cell markers identified in a recent study from our laboratory⁷. In that study, we microdissected the subependymal zone and performed single-cell RNA sequencing to define a set of genes—FOXJ1, TMEM212, DYNLRB2, CCDC153, and PIFO—that are exclusively expressed by ependymal cells (panel a, b, c). We deliberately excluded CRYAB, CXCL12, CCN1, CCN2, and S100A11 from the marker list due to their lower specificity (panel b, c). When examining the expression of the high-specific markers (panel d) in our human astrocyte subset (original Extended Data Fig. 4), we detected a minor contamination of ependymal cells, as indicated by the ependymal cell score in cluster 6 (panel e). We then repeated the full set analyses shown in Extended Data Figure 4 after removing this contamination. As shown in the revised Extended Data Fig. 4, the results remained unchanged.

Exclusion of ependymal cells. **a**, Left: Schematic representation of the three regions dissected from 2–3-month-old C57BL/6J mice to generate single-cell suspensions for scRNA-seq. Middle: t-SNE plot of the scRNA-seq data, with cells color-coded by dissection region. Right: Same t-SNE plot with cells color-coded by cell type. **b**, Expression levels of ependymal cell markers across cell types. **c**, t-SNE plots illustrating expression levels of ependymal cell markers across cell types revealing their specificity. **d**, UMAP plots of the original human dataset illustrating expression of selected ependymal cell markers. **e**, UMAP plots of the original human dataset depicting the Ependymal cell score demarcating ependymal cells (left). Cluster 6 (right) was removed from the dataset.

References:

1 Prigge, J. R. *et al.* Nuclear double-fluorescent reporter for in vivo and ex vivo analyses of biological
2 transitions in mouse nuclei. *Mamm Genome*, doi:10.1007/s00335-013-9469-8 (2013).

3

4

[Redacted]

5

6

7

Bocchi, R. *et al.* Astrocyte heterogeneity reveals region-specific astrogenesis in the white matter. *Nat
Neurosci* **28**, 457-469, doi:10.1038/s41593-025-01878-6 (2025).

Zhou et al, manuscript

Point-by-point answer to the Reviewers

Reviewer #4:

Prior to reviewing the revised manuscript, I would like to provide feedback regarding the authors' responses to my previous comments.

R4C1. The authors must provide definitive evidence that Emx1-Cre does not exhibit ubiquitous expression in all dorsal cortical radial glia.

Despite being emphasized twice in previous reviews, this critical issue has not been properly addressed in the authors' response.

In the seminal Emx1-Cre paper by Gorski et al. (2002) titled "Cortical excitatory neurons and glia, but not GABAergic neurons, are produced in the Emx1-expressing lineage" (J Neurosci. 22(15): 6309-14, PMID: 12151506), the authors clearly demonstrated that Emx1-Cre labels all pyramidal neurons and astrocytes in the dorsal cortex. This paper has been widely cited over 1,500 times. This finding, supported by the supplemental figure (attached) depicting recombination in the adult Emx1-cre; Z/AP brain. Specifically, panel (N) of the supplemental figure shows an X-gal-stained 40 μm coronal section of an adult Emx1-cre; Z/AP brain, revealing extensive recombination throughout the cortex and hippocampus. Notably, X-gal-positive staining was observed in pyramidal neurons and astrocytes.

[Redacted]

Response R4C1. While we acknowledge that the Gorski et al. (2002) study demonstrated broad labeling of cortical astrocytes using Emx1-Cre, we respectfully point out that the Z/AP reporter used in that study does not allow for single-cell resolution, and the data presented—particularly in Fig. 4J–L—relies on *in vitro* co-staining with S100b without providing quantification. Therefore, although Emx1⁺ RGCs clearly contributes to cortical astrocytes, these results do not preclude the existence of astrocytes derived from Emx1⁻ progenitors.

Our study confirms the contribution of Emx1⁺ RGCs to cortical astrocytes, consistent with Gorski et al. Crucially, however, we go further by employing a multi-modal strategy—including single-cell RNA sequencing, lineage tracing (TrackerSeq), spatial transcriptomics, and *in vivo* fate mapping using *Emx1::Cre; ROSA^{nT-nG}* (a dual nuclear reporter) mice—to provide quantitative evidence that approximately 25–30% of cortical astrocytes are not labeled by Emx1::Cre and express Olig2 (Fig. 3m–n; Supplementary Fig. 8g–i). This robust finding reveals the existence of a distinct astrocyte lineage originating from Emx1⁻ RGCs.

[Redacted]

R4C2. The authors propose a highly unconventional model in their discussion and conclusion Fig. 5g, positing the existence of two distinct radial glial populations:

The author now claimed that in the abstract: The first lineage derives from Emx1+ radial glial cells that initially generate neurons and later switch to astrocyte production. The second lineage (derives from Emx1-/low radial glial cells), with minimal neuronal output, predominantly produces a distinct subset of astrocytes marked by Olig2.

Given that nearly all cortical pyramidal neurons and astrocytes originate from Emx1-Cre-expressing radial glia (as established in prior literatures and my earlier comments), the authors' conclusion is untenable. Without strong direct evidence demonstrating the existence of Emx1-negative radial glia contributing to cortical astrocytes, this claim remains unsupported.

Response R4C2. Our conclusion that two distinct radial glial populations (Emx1⁺ and Emx1⁻) contribute to cortical astrocytes is supported by multiple, independent lines of evidence:

1. Single-cell clonal tracing (TrackerSeq) reveals the existence of astrocyte-only clones, particularly enriched in the Olig2 lineage, with minimal to no contribution to excitatory neurons. This suggests the presence of a gliogenic RGC population with limited neurogenic potential.
2. Fate mapping using *Emx1::Cre; ROSA^{nl-nG}* mice shows that ~25–30% of cortical astrocytes are not labeled by the Emx1::Cre driver and express Olig2 (Fig. 3m–n; Supplementary Fig. 8g–i), supporting the existence of a population of astrocytes derived from Emx1⁻ RGCs.
3. Transcriptomic analysis at E18.5 identifies two transcriptionally distinct RGC subtypes—RGC_1 and RGC_2—with clear lineage segregation: RGC_1 gives rise to neurons and S100a11-lineage astrocytes, while RGC_2 generates Olig2-lineage astrocytes almost exclusively (Fig. 3c).
4. Spatial transcriptomics at E14.5 confirms the coexistence of two spatially distinct RGC populations in the dorsal VZ, marked by non-overlapping gene expression profiles (Fig. 3k–l).
5. **[Redacted]**

R4C3. The authors' interpretation in Extended Data Fig. 4 directly contradicts established findings from Trevino et al. (Cell, 2021; PMID: 34390642), which showed that S100A11 and KLF2 are expressed mainly in human cortical ependymal cells rather than astrocytes.

In their response, the authors once again failed to address my fundamental question. Specifically, the newly revised Extended Data Fig. 4 does not demonstrate the presence of S100a11-expressing astrocytes in the human cortex.

Response R4C3. We apologize for the confusion. Gene expression can differ between species, which is why our analysis in Supplementary Fig. 4a–d is based on a set of 17 mouse trajectory-specific markers (ADAMTS1, AHNAK, ANXA5, ATP1A1, C1QL3, C4B, DYNLRB2, FOXJ1, IFITM3, IQCG, MNS1, PHLDA1, S100A11, S100A4, SEPTIN3, TAGLN2, ARX). Using this gene set, we demonstrate that the developing human cortex contains astrocytes enriched for the S100a11 trajectory gene set. Although S100A11 itself is not highly expressed in human astrocytes, our analysis focuses on trajectory scores and pseudotime progression from RGCs to astrocytes, rather than on S100A11 expression alone. These findings support the presence of both astrocyte lineages in the human cortex.

R4C4. Furthermore, I note significant issues with the manuscript's scholarly rigor. On page 13, line 412 of the main text, key references supporting their argument have been omitted in this revision, despite being present in previous versions.

Response R4C4. Thank you for bringing this to our attention. We apologize for the oversight—this was a mistake on our part. We have now restored the previously cited references supporting our interpretation at the indicated location in the revised manuscript. In addition, we have thoroughly checked the entire text to ensure that no other essential references were inadvertently removed during revision.

R4C5. Taken together, these critical flaws lead me to conclude that the paper's major conclusions are fundamentally unsupported. The cumulative evidence suggests these findings cannot be accepted as scientifically valid.

Response R4C5. We respectfully disagree with the reviewer's assertion that the major conclusions are unsupported. Our conclusions are built upon converging evidence from transcriptomics, spatial data, lineage tracing, and genetic perturbations. Nonetheless, we have carefully revised the abstract, discussion, and conclusions to moderate the claims and emphasize that our findings support—not definitively prove—the existence of two progenitor pools contributing to cortical astrocytes.